# Targeting DNA topoisomerases or checkpoint kinases results in an overload of chaperone systems, triggering aggregation of a metastable subproteome

Wouter Huiting[1], Suzanne L Dekker[1†], Joris CJ van der Lienden[1†], Rafaella Mergener[1], Maiara K Musskopf[1], Gabriel V Furtado[1], Emma Gerrits[1], David Coit[2], Mehrnoosh Oghbaie[2,3], Luciano H Di Stefano[3], Hein Schepers[1], Maria AWH van Waarde-Verhagen[1], Suzanne Couzijn[1], Lara Barazzuol[1,4], John LaCava[2,3*‡], Harm H Kampinga[1], Steven Bergink[1*]

[1]Department of Biomedical Sciences of Cells and Systems, University Medical Center Groningen, University of Groningen, Groningen, Netherlands; [2]Laboratory of Cellular and Structural Biology, The Rockefeller University, New York, United States; [3]European Research Institute for the Biology of Ageing, University Medical Center Groningen, University of Groningen, Groningen, Netherlands; [4]Department of Radiation Oncology, University Medical Center Groningen, University of Groningen, Groningen, Netherlands

*For correspondence:
j.p.lacava@umcg.nl (JL);
s.bergink@rug.nl (SB)

‡Technical proteomics correspondence to: j.p.lacava@umcg.nl

†These authors contributed equally to this work

**Abstract** A loss of the checkpoint kinase ataxia telangiectasia mutated (ATM) leads to impairments in the DNA damage response, and in humans causes cerebellar neurodegeneration, and an increased risk of cancer. A loss of ATM is also associated with increased protein aggregation. The relevance and characteristics of this aggregation are still incompletely understood. Moreover, it is unclear to what extent other genotoxic conditions can trigger protein aggregation as well. Here, we show that targeting ATM, but also ATR or DNA topoisomerases, results in the widespread aggregation of a metastable, disease-associated subfraction of the proteome. Aggregation-prone model substrates, including Huntingtin exon 1 containing an expanded polyglutamine repeat, aggregate faster under these conditions. This increased aggregation results from an overload of chaperone systems, which lowers the cell-intrinsic threshold for proteins to aggregate. In line with this, we find that inhibition of the HSP70 chaperone system further exacerbates the increased protein aggregation. Moreover, we identify the molecular chaperone HSPB5 as a cell-specific suppressor of it. Our findings reveal that various genotoxic conditions trigger widespread protein aggregation in a manner that is highly reminiscent of the aggregation occurring in situations of proteotoxic stress and in proteinopathies.

## Editor's evaluation

This study investigates how different sources of genotoxic stress exacerbate proteome instability, leading to the formation of protein aggregates. These effects are in part caused by an impaired chaperone network and can be relieved by the action of small chaperones with anti-amyloidogenic function. This work functionally connects two fields of research: responses to genotoxic stress and protein aggregation, both of which have fundamental implications for cellular stability and

homeostasis during aging. Connecting these two fields sheds new light on basic mechanisms of cell stability and has the potential to help design interventions that buffer both DNA damage responses and proteome stability.

## Introduction

The PI3K-like serine/threonine checkpoint kinase ataxia telangiectasia mutated (ATM) functions as a central regulator of the DNA damage response (DDR) and is recruited early to DNA double-strand breaks (DSBs) by the MRE11/RAD50/NBS1 (MRN) complex (*Shiloh and Ziv, 2013*). Defects in ATM give rise to ataxia-telangiectasia (A-T), a multisystem disorder that is characterized by a predisposition to cancer and progressive neurodegeneration (*McKinnon, 2012*).

Impaired function of ATM has also been linked to a disruption of protein homeostasis and increased protein aggregation (*Corcoles-Saez et al., 2018*; *Lee et al., 2018*; *Liu et al., 2005*). Protein homeostasis is normally maintained by protein quality control systems, including chaperones and proteolytic pathways (*Hipp et al., 2019*; *Labbadia and Morimoto, 2015*). Together, these systems guard the balance of the proteome by facilitating correct protein folding, providing conformational maintenance, and ensuring timely degradation. When the capacity of protein quality control systems becomes overwhelmed during (chronic) proteotoxic stress, the stability of the proteome can no longer be sufficiently guarded, causing proteins to succumb to aggregation more readily. Proteins that are expressed at a relatively high level compared to their intrinsic aggregation propensity, a state referred to as 'supersaturation,' have been shown to be particularly vulnerable in this respect (*Ciryam et al., 2015*). A loss of protein homeostasis and the accompanying widespread aggregation can have profound consequences, and is associated with a range of (degenerative) diseases, including neuro-degeneration (*Kampinga and Berginka, 2016*; *Klaips et al., 2018*; *Ross and Poirier, 2004*).

The characteristics and relevance of the aggregation induced by a loss of ATM are still largely unclear. Loss of MRE11 has recently also been found to result in protein aggregation (*Lee et al., 2021*), and since MRE11 and ATM function in the same DDR pathway, this raises the question whether other genotoxic conditions can challenge protein homeostasis as well (*Ainslie et al., 2021Huiting and Bergink, 2020*).

Here, we report that not just impaired function of ATM, but also inhibition of the related check-point kinase ataxia telangiectasia and Rad3-related (ATR), as well as chemical trapping of topoisomer-ases (TOPs) using chemotherapeutic TOP poisons leads to widespread protein aggregation. Through proteomic profiling, we uncover that the increased protein aggregation induced by these genotoxic conditions overlaps strongly with the aggregation observed under conditions of (chronic) stress and in various neurodegenerative disorders, both in identity and in biochemical characteristics. In addition, we find that these conditions accelerate the aggregation of aggregation-prone model substrates, including the Huntington's disease-related polyglutamine exon 1 fragment. We show that the widespread protein aggregation is the result of an overload of protein quality control systems, which cannot be explained by any quantitative changes in the aggregating proteins or by genetic alterations in their coding regions. This overload forces a shift in the equilibrium of protein homeostasis, causing proteins that are normally kept soluble by chaperones to now aggregate. Which proteins succumb to aggregation depends on the ground state of protein homeostasis, including the wiring of chaperone systems in that cell. Finally, we provide evidence that the protein aggregation induced by genotoxic stress conditions is amenable to modulation by chaperone systems: whereas inhibition of HSP70 exac-erbates aggregation, we also provide a proof of concept that aggregation can be rescued in a cell line-specific manner by increasing the levels of the small heat shock protein HSPB5 (αB-crystallin).

## Results

### Protein aggregation is increased upon targeting ATM, ATR, or DNA TOPs

Aggregated proteins are often resistant to solubilization by SDS, and they can therefore be isolated using a step-wise detergent fractionation and centrifugation method. We isolated 1% SDS-resistant proteins (from here on referred to as aggregated proteins) and quantified these by SDS-PAGE followed

**eLife digest** Cells are constantly perceiving and responding to changes in their surroundings, and challenging conditions such as extreme heat or toxic chemicals can put cells under stress. When this happens, protein production can be affected. Proteins are long chains of chemical building blocks called amino acids, and they can only perform their roles if they fold into the right shape. Some proteins fold easily and remain folded, but others can be unstable and often become misfolded. Unfolded proteins can become a problem because they stick to each other, forming large clumps called aggregates that can interfere with the normal activity of cells, causing damage.

The causes of stress that have a direct effect on protein folding are called proteotoxic stresses, and include, for example, high temperatures, which make proteins more flexible and unstable, increasing their chances of becoming unfolded. To prevent proteins becoming misfolded, cells can make 'protein chaperones', a type of proteins that help other proteins fold correctly and stay folded. The production of protein chaperones often increases in response to proteotoxic stress. However, there are other types of stress too, such as genotoxic stress, which damages DNA. It is unclear what effect genotoxic stress has on protein folding.

Huiting et al. studied protein folding during genotoxic stress in human cells grown in the lab. Stress was induced by either blocking the proteins that repair DNA or by 'trapping' the proteins that release DNA tension, both of which result in DNA damage. The analysis showed that, similar to the effects of proteotoxic stress, genotoxic stress increased the number of proteins that aggregate, although certain proteins formed aggregates even without stress, particularly if they were common and relatively unstable proteins.

Huiting et al.'s results suggest that aggregation increases in cells under genotoxic stress because the cells fail to produce enough chaperones to effectively fold all the proteins that need it. Indeed, Huiting et al. showed that aggregates contain many proteins that rely on chaperones, and that increasing the number of chaperones in stressed cells reduced protein aggregation.

This work shows that genotoxic stress can affect protein folding by limiting the availability of chaperones, which increases protein aggregation. Remarkably, there is a substantial overlap between proteins that aggregate in diseases that affect the brain – such as Alzheimer's disease – and proteins that aggregate after genotoxic stress. Therefore, further research could focus on determining whether genotoxic stress is involved in the progression of these neurological diseases.

by in-gel protein staining. In line with previous findings (*Lee et al., 2018*), we find that knocking out *ATM* in both U2OS and HEK293 results in an increase in protein aggregation (*Figure 1A and B*, *Figure 1—figure supplement 1A–C*). Transient chemical inhibition of ATM (48–72 hr prior to fractionation; *Figure 1—figure supplement 1D*) resulted in an increase in aggregated proteins in HEK293T cells as well (*Figure 1C and D*).

Using the same experimental set-up, we examined the impact on aggregation of targeting other DDR components. This revealed that chemical inhibition of the checkpoint signaling kinase ATR also enhanced protein aggregation (*Figure 1C and D*). Inhibition of tyrosyl-DNA-phosphodiesterase 1 (TDP1), which repairs various 3'-blocking lesions including topoisomerase 1 (TOP1) cleavage complexes, had no clear effect on protein aggregation (*Figure 1C and D*). This could be a result of functional redundancy or limited TOP1 trapping occurring under unstressed conditions in a timeframe of 72 hr. We therefore also directly targeted TOPs using the chemotherapeutic compounds camptothecin (CPT) and etoposide (Etop). The genotoxic impact of CPT and Etop is a well-documented consequence of their ability to trap (i.e., 'poison') respectively TOP1 and TOP2 cleavage complexes on the DNA, resulting in DNA damage (*Pommier et al., 2010*). Strikingly, we found that transient treatment with either compound caused a particularly strong increase in protein aggregation (*Figure 1C and D*), which was dose-dependent (*Figure 1E and F*). Treatment of U2OS cells with CPT led to a dose-dependent increase in aggregation as well, although at higher doses compared to HEK293T cells (*Figure 1—figure supplement 1E*). Inhibition of poly(ADP-ribose)polymerases 1–3 (PARP1-3), involved in single-strand break repair, did not increase aggregation (*Figure 1C and D*). Recently, it was reported that PARP inhibition reduces the enhanced aggregation triggered by a loss of ATM (*Lee et al., 2021*), something we find as well for CPT-treated cells (*Figure 1—figure supplement 1F and G*).

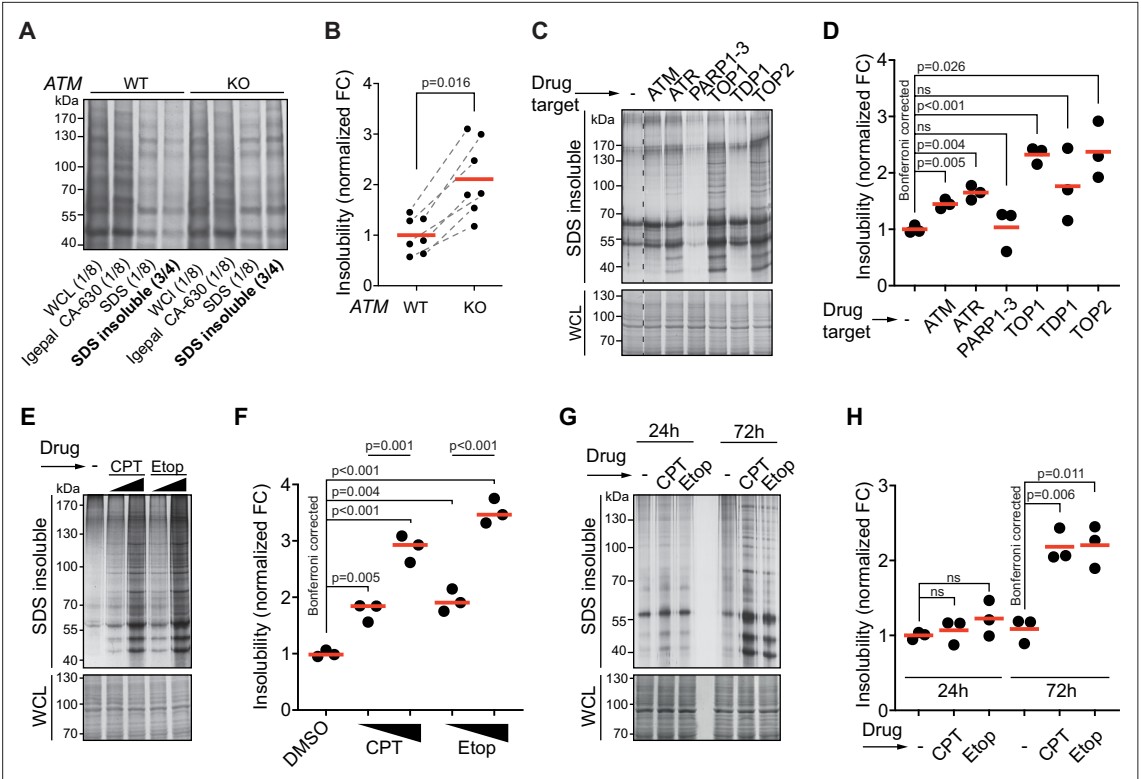

**Figure 1.** Protein aggregation is increased following a functional loss of ataxia telangiectasia mutated (ATM), ataxia telangiectasia and Rad3-related (ATR), and upon topoisomerase poisoning. See also *Figure 1—figure supplement 1*. (**A**) In-gel Coomassie staining of indicated fractions of cell extracts of WT and *ATM* KO U2OS cells. The relative amounts of each fraction loaded are indicated. (**B**) Quantification of (**A**). Circles depict individual experiments; gray dotted lines depict matched pairs. Wilcoxon matched-pairs signed-rank test. (**C**) Aggregated (silver stain) and whole-cell lysate (WCL; Coomassie) fractions of HEK293T cells treated transiently with chemical agents targeting the indicating proteins (see *Table 1* for drugs and doses used; for etoposide [Etop]: 3 μM; for camptothecin [CPT]: 100 nM). See also *Figure 1—figure supplement 1D*. (**D**) Quantification of (**C**). Circles depict individual experiments. Two-tailed Student's *t*-test with Bonferroni correction. (**E**) Protein fractions of HEK293T cells treated transiently with increasing amounts of CPT (20–100 nM) or Etop (0.6–3 μM). (**F**) Quantification of (**E**). Two-tailed Student's *t*-test with Bonferroni correction. (**G**) Protein fractions of HEK293T cells treated transiently with CPT (40 nM) or Etop (1.5 μM), targeting TOP1 or TOP2, respectively, 24 hr or 72 hr after treatment. (**H**) Quantification of (**G**). Two-tailed Student's *t*-test with Bonferroni correction. In (**B**), (**D**), (**F**), and (**H**), the red line indicates the mean.

The online version of this article includes the following source data and figure supplement(s) for figure 1:

**Source data 1.** Data from *Figure 1A*.

**Source data 2.** Data from *Figure 1C*.

**Source data 3.** Data from *Figure 1E*.

**Source data 4.** Data from *Figure 1G*.

**Figure supplement 1.** Aggregation is increased in cells lacking ataxia telangiectasia mutated (ATM).

**Figure supplement 1—source data 1.** Data from *Figure 1—figure supplement 1A*.

**Figure supplement 1—source data 2.** Data from *Figure 1—figure supplement 1B*.

**Figure supplement 1—source data 3.** Data from *Figure 1—figure supplement 1E*.

**Figure supplement 1—source data 4.** Data from *Figure 1—figure supplement 1F*.

Neither CPT nor Etop treatment in HEK293T cells had any effect on aggregation within the first 24 hr (*Figure 1G and H*). This reveals that the increased aggregation occurs only late and argues that it does not stem from any immediate, unknown damaging effect of either CPT or Etop on mRNA or protein molecules. Together, these data indicate that the increased protein aggregation triggered by targeting ATM, ATR, and TOPs is a late consequence of genotoxic stress.

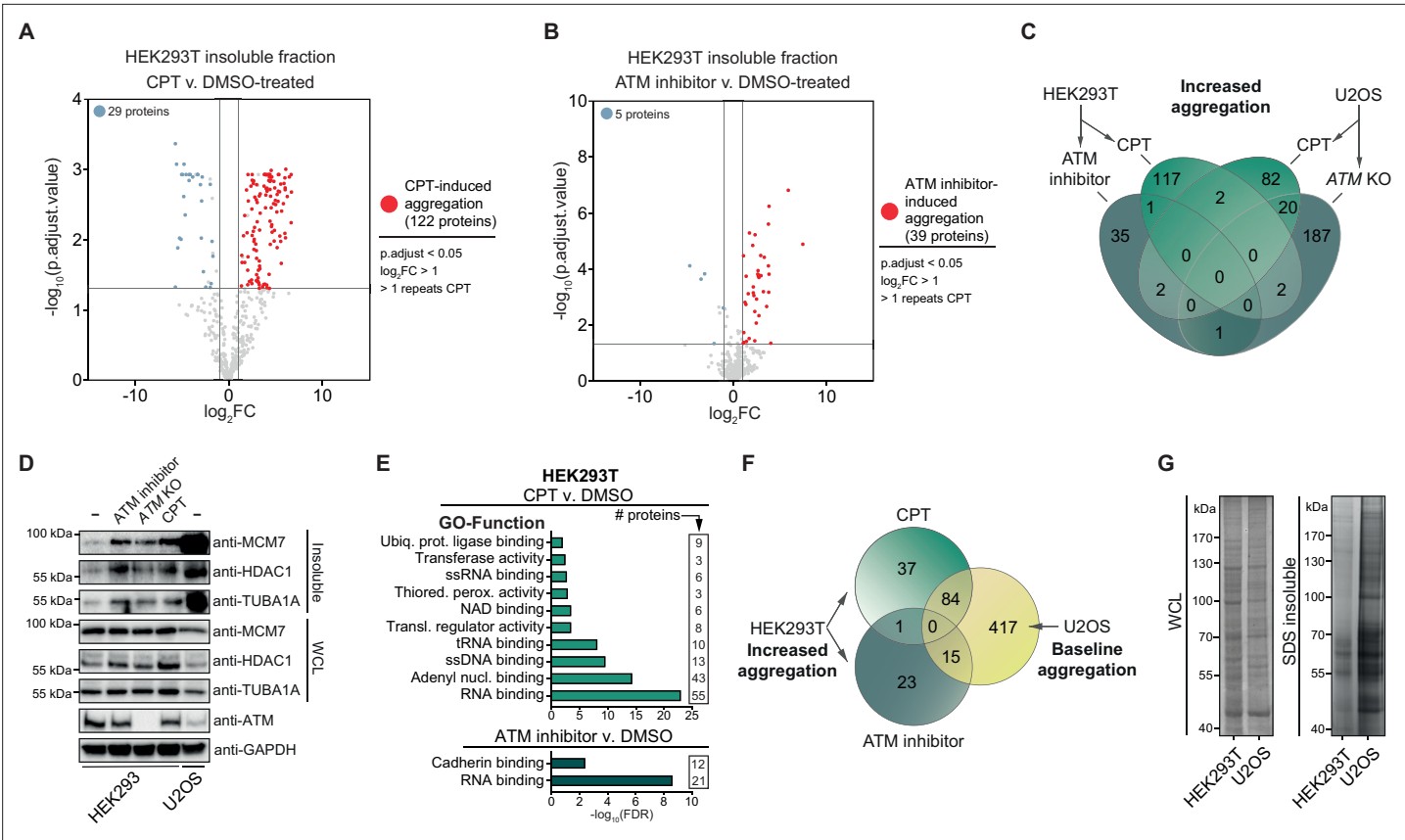

**Figure 2.** Camptothecin (CPT) and ataxia telangiectasia mutated (ATM) loss drives aggregation in a cell-type-dependent manner. See also *Figure 2—figure supplement 1*. (**A**) Volcano plot of label-free quantification (LFQ) MS/MS analysis of the aggregated fractions of DMSO and CPT-treated HEK293T cells. n = 4. Only proteins identified in >1 repeats of either case or control are shown. (**B**) Volcano plot of LFQ MS/MS analysis of the aggregated fractions of DMSO and ATM inhibitor-treated HEK293T cells. n = 4. Only proteins identified in >1 repeats of either case or control are shown. (**C**) Venn diagram showing overlap between U2OS and HEK293T increased aggregation, after the indicated treatments. (**D**) Western blot using the indicated antibodies on the aggregated and whole-cell lysate (WCL) fractions of drug-treated and *ATM* KO HEK293 cells, and wild-type U2OS cells. n = 2. (**E**) GO term analysis (Function) of the increased aggregation in CPT- or ATM-inhibitor-treated HEK293T cells. (**F**) Venn diagram showing overlap between increased aggregation after the indicated treatments in HEK293T cells and baseline aggregation in U2OS cells. (**G**) Aggregated (silver stain) and WCL (Coomassie) fractions of untreated HEK293T and U2OS cells. n = 2.

The online version of this article includes the following source data and figure supplement(s) for figure 2:

**Source data 1.** Data from *Figure 2D*.

**Source data 2.** Data from *Figure 2G*.

**Figure supplement 1.** GO term analyses of the aggregation triggered by camptothecin (CPT) and ataxia telangiectasia mutated (ATM) loss.

## CPT and ATM loss drives aggregation in a cell-type-dependent manner

To investigate the nature of the proteins that become aggregated after genotoxic stress, we subjected the SDS-insoluble protein aggregate fractions and whole-cell lysates (WCL) of control (DMSO) and CPT-treated HEK293T cells to label-free proteomics (*Figure 2—figure supplement 1A*). Using a stringent cutoff (Benjamini–Hochberg corrected p<0.05; –1>log$_2$fold change >1; identified in >1 repeats of CPT-treated cells), we determined that 122 proteins aggregated significantly more after CPT treatment compared to only 29 proteins that aggregated less (*Figure 2A*, *Supplementary file 1*). These 122 proteins aggregate highly consistent (*Supplementary file 1*). Most of them were not identified as aggregating in untreated cells, implying that they are soluble under normal conditions.

We next used the same MS/MS approach to investigate the aggregation triggered by inhibition of ATM in HEK293T cells. We detected 39 proteins that aggregated significantly more in ATM-inhibited cells compared to control cells and 5 proteins that aggregated less (*Figure 2B*, *Supplementary file 1*). Surprisingly, only one protein was found to aggregate more after both CPT treatment and

inhibition of ATM (*Figure 2C*), suggesting that these genotoxic conditions drive the aggregation of different proteins. However, we noted that several proteins that aggregated more after CPT treatment were absent in the dataset of ATM inhibition (*Supplementary file 1*), suggesting that we may have only identified the most abundantly aggregating proteins. We selected MCM7, TUBA1A, and HDAC1, three proteins that were identified in our MS/MS analysis, to consistently aggregate more in CPT-treated HEK293T cells but that were not picked up in the ATM inhibition MS/MS analysis and confirmed that all three aggregated more in CPT-treated HEK293 cells (*Figure 2D*). MCM7, TUBA1A, and HDAC1 also aggregated more than in unstressed conditions after treatment of HEK293 cells with ATM inhibitor or when *ATM* was knocked out completely (*Figure 2D*). These findings indicate that these different genotoxic conditions drive aggregation similarly, although the most prominent aggregating proteins differ.

Next, we investigated the aggregation caused by a loss of ATM or CPT treatment in U2OS cells, a cell line that has been used previously as well to study the effect of a loss of ATM on protein aggregation. We found 210 proteins that aggregated more in *ATM* KO cells, while 53 proteins aggregated less (*Figure 2—figure supplement 1B*, *Supplementary file 1*). Of these 210 proteins, 114 were also found to aggregate more in *ATM*-depleted U2OS cells in a recent study by *Lee et al., 2021*. Treatment of U2OS cells with CPT resulted in 106 proteins aggregating more and 61 proteins to aggregate less (*Figure 2—figure supplement 1C*, *Supplementary file 1*). Close to 20% of proteins that aggregated more after CPT treatment also aggregated in *ATM* KO U2OS cells (20/106) (*Figure 2C*). Similar to the induced aggregation in HEK293T cells, proteins that aggregate more in U2OS *ATM* KO or CPT-treated cells appear to be largely soluble in wild-type cells, but now aggregate consistently (*Supplementary file 1*).

At first glance, protein aggregation caused by a (functional) loss of ATM or CPT treatment in U2OS cells seemed to be quite different from the aggregation observed in HEK293T cells. Not a single protein was found to aggregate more in all four different conditions, only two proteins aggregated more in both HEK293T and U2OS cells after CPT treatment, and only one protein overlapped between *ATM* KO U2OS cells and ATM-inhibited HEK293T cells (*Figure 2C*). Interestingly, a GO term analysis of the aggregating proteomes did reveal overlap across the different treatments and the two cell lines (*Figure 2E*, *Figure 2—figure supplement 1D–I*). Cytoskeleton-related terms, including microtubule and microfibril, are enriched across the different aggregating proteomes (*Figure 2—figure supplement 1D–F, H*). However, enrichment of most GO terms is restricted to a specific treatment and/or cell line. For example, proteins that aggregated more in CPT-treated HEK293T cells are enriched for nucleotide binding terms, most prominently RNA binding, which is highly enriched among proteins that aggregate after ATM inhibition as well (*Figure 2E*). Specifically CPT treatment in HEK293T cells drives the aggregation of mitochondrial components (*Figure 2—figure supplement 1D*). In U2OS cells, proteins that aggregate more after a loss of ATM or CPT treatment appear to be enriched for components involved in cell-cell contact, including cell adhesion and cellular membrane processes (*Figure 2—figure supplement 1F–H*). .

As protein aggregation can manifest vastly different in distinct cell types (*David et al., 2010*; *Freer et al., 2016*), we examined which proteins aggregated consistently in HEK293T and U2OS cells, regardless of the presence or absence of genotoxic stress. Importantly, within each cell line, these 'consistently' aggregating proteins show a very high overlap between experiments (~80% overlap, see also *Supplementary file 1*). Based on this, we defined a 'baseline aggregating fraction' for each cell line. This consisted of aggregating proteins that were not changed upon the genotoxic treatments: these proteins were detected in at least two experimental replicates of both treated and untreated cells and exhibited p-adjusted values of >0.05 in t-test comparisons, consistent with no significant effect (*Supplementary file 1*). This revealed that 66% (118/179) of the HEK293T baseline fraction aggregates in the U2OS baseline as well (*Figure 2—figure supplement 1J*). Importantly, 62% (99/160) of the proteins that aggregated more in CPT- or ATM inhibitor-treated HEK293T cells are also part of the U2OS baseline (*Figure 2F*). This indicates that in U2OS cells afar bigger cluster of proteins ends up in aggregates, even under normal conditions. Indeed, silver staining revealed that in unstressed U2OS cells protein aggregation is substantially more prominent than in untreated HEK293T cells (*Figure 2G*). This is also reflected in MCM7, TUBA1A, and HDAC1, all three of which aggregate strongly already in (untreated) wild-type U2OS cells (*Figure 2D*). These findings indicate that the lack of overlap between proteins that aggregate after CPT- or ATM inhibitor treatment in

HEK293T and proteins that aggregate in U2OS after CPT treatment or in ATM KO cells is primarily a reflection of a different proteome and a different background aggregation in these two cell lines.

## Proteins that aggregate after genotoxic stress represent a metastable subproteome

These data indicate that the genotoxic conditions of TOP1 poisoning and ATM loss have a cell line-dependent impact on protein aggregation. In both HEK293T and U2OS cells, protein aggregation does not appear to be limited to a specific location or function but affects proteins throughout the proteome. This suggests that the aggregation is primarily driven by the physicochemical characteristics of the proteins involved.

A key determinant of aggregation is supersaturation. Protein supersaturation refers to proteins that are expressed at high levels relative to their intrinsic propensity to aggregate, which makes them vulnerable to aggregation. Supersaturation has been shown to underlie the widespread protein aggregation observed in age-related neurodegenerative diseases, and in general aging (*Ciryam et al., 2015*; *Ciryam et al., 2019*; *Freer et al., 2019*; *Kundra et al., 2017*; *Noji et al., 2021*). The relevance of supersaturation is underlined by the notion that evolutionary pressures appear to have shaped proteomes along its lines, so that at a global level protein abundance is inversely correlated with aggregation propensity (*Tartaglia et al., 2007*). To determine the role of protein supersaturation in the aggregation observed in our experiments, we first defined a control group of proteins that were *not* identified as aggregating (NIA) for HEK293T cells to serve as a benchmark. This group consisted of all proteins that were only identified in the HEK293T WCL, and not in the SDS-insoluble fractions (see also *Supplementary file 1*). We next examined the intrinsic aggregation propensities of proteins using the aggregation prediction tools TANGO (*Fernandez-Escamilla et al., 2004*) and CamSol (*Sormanni and Vendruscolo, 2019*). Surprisingly, we found that aggregated proteins have in general a slightly lower (for the baseline aggregation) or equal (for CPT- and ATM inhibitor-induced aggregation) intrinsic propensity to aggregate compared to NIA proteins (*Figure 3A*, *Figure 3—figure supplement 1A*). However, even proteins with a low intrinsic propensity to aggregate can be supersaturated and be vulnerable to aggregation, when they are expressed at sufficiently high levels. Interestingly, our MS/MS analysis revealed that proteins that aggregate in CPT- or ATM-inhibited-treated HEK293T cells are in general highly abundant compared to NIA proteins in (*Figure 3B*). Cross-referencing the aggregated proteins in our datasets against a cell-line-specific NSAF reference proteome (*Geiger et al., 2012*) confirmed this (*Figure 3—figure supplement 1B*). After performing RNA sequencing on the same HEK293T cell samples that we used for our MS/MS analysis (*Figure 2—figure supplement 1A*, *Supplementary file 2*), we found that genes coding for the aggregating proteins are in general higher expressed than genes coding for NIA proteins (*Figure 3C*).

To evaluate whether these proteins are indeed supersaturated, we used the method validated by Ciryam et al., which uses transcript abundance and aggregation propensity as predicted by TANGO to estimate supersaturation (*Ciryam et al., 2013*). Using the RNA-sequencing data (*Supplementary file 2*), we confirmed that aggregating proteins are in general indeed more supersaturated than NIA proteins (*Figure 3D–F*). Cross-referencing our data against the composite human supersaturation database generated by Ciryam et al. yielded a similar picture (*Figure 3—figure supplement 1C*).

Although the relative supersaturation of aggregating proteins in HEK293T cells is intriguing, our data also indicates that most supersaturated proteins did not become SDS-insoluble, even after treatment with CPT or ATM inhibition (*Figure 3F*). Supersaturation only relates to overall protein concentration per cell, but within a cell, local protein concentrations can differ. A prime example of this is the partitioning of proteins in so-called biomolecular condensates through liquid-liquid phase separation (LLPS). LLPS can increase the local concentration of proteins, which has been shown to be important for a wide range of cellular processes (*Lyon et al., 2021*). However, it also comes with a risk of transitioning from a liquid to a solid, and even amyloid state. Indeed, a large amount of recent data have clearly demonstrated that proteins that engage in LLPS are overrepresented among proteins that aggregate in various proteinopathies (reviewed in *Alberti and Hyman, 2021*). Using catGRANULE (*Mitchell et al., 2013*; http://tartaglialab.com), we find that HEK293T baseline aggregation and both CPT- and ATM inhibitor-induced aggregation are indeed made up of proteins that have a higher average LLPS propensity than NIA proteins (*Figure 3G*). HEK293T baseline and ATM inhibitor-induced aggregation are also enriched for proteins that have a high propensity to engage in LLPS-relevant

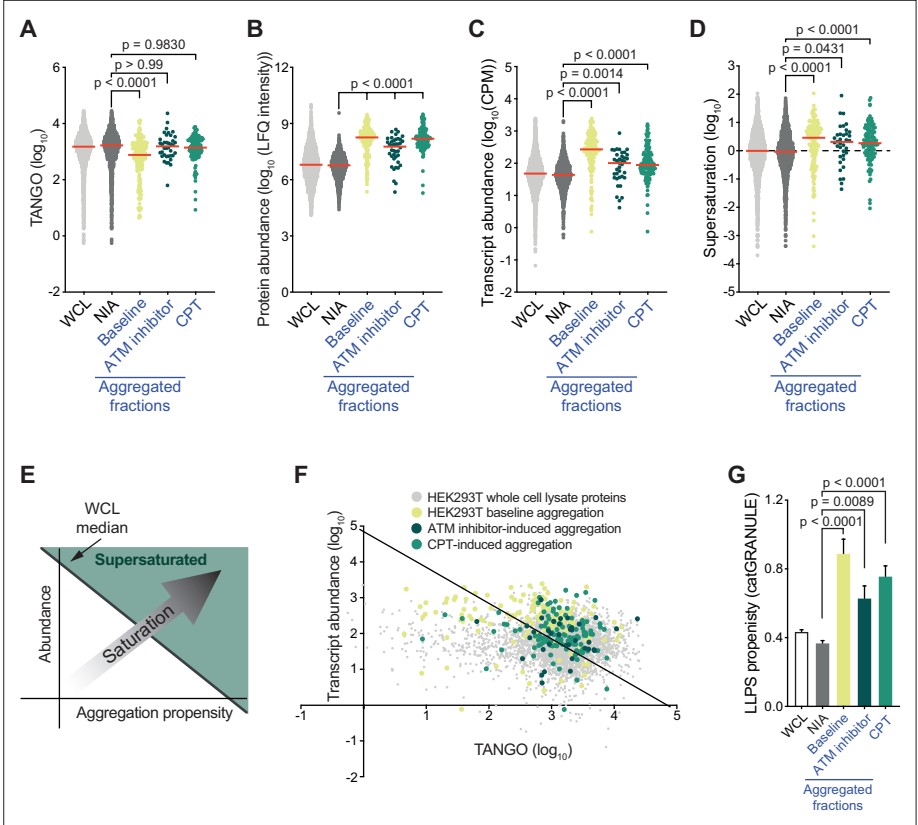

**Figure 3.** Proteins that aggregate after topoisomerase I poisoning are supersaturated and prone to engage in liquid-liquid phase separation (LLPS). See also *Figure 3—figure supplements 1 and 2*. (**A**) TANGO scores of HEK293T whole-cell lysate (WCL), nonaggregated proteins (NIA), and aggregated fractions. (**B**) Protein abundance of HEK293T WCL, nonaggregated proteins (NIA), and aggregated fractions as measured by label-free quantification (LFQ) intensities. (**C**) Transcript abundances of HEK293T WCL, nonaggregated proteins (NIA), and aggregated fractions (as measured by RNAseq). (**D**) Supersaturation scores of HEK293T WCL, nonaggregated proteins (NIA), and aggregated fractions. (**E**) Clarification of (**F**). (**F**) Transcript abundances (as measured by RNAseq) plotted against TANGO scores for the complete HEK293T MS/MS analysis. Proteins above the diagonal (=HEK293T median saturation score, calculated using the HEK293T WCL dataset) are relatively supersaturated. (**G**) catGRANULE scores for the indicated protein fractions in HEK293T cells. In all graphs, individual proteins and median values (red lines) are shown. p-Values are obtained by Kruskal–Wallis tests followed by Dunn's correction for multiple comparisons.

The online version of this article includes the following figure supplement(s) for figure 3:

**Figure supplement 1.** Proteins that aggregate after camptothecin treatment and ataxia telangiectasia mutated (ATM) loss represent a vulnerable subfraction of the proteome.

**Figure supplement 2.** GO term analyses of RNAseq data.

pi-pi interactions, as indicated by both a higher average PScore and a larger percentage of proteins that have a PScore > 4 (i.e., above the threshold defined by *Vernon et al., 2018*; *Figure 3—figure supplement 1D and E*). Inversely, dividing NIA proteins into supersaturated and non-supersaturated subgroups reveals that they have a similarly low average LLPS propensity (*Figure 3—figure supplement 1F and G*). This points out that a high LLPS propensity can discriminate supersaturated proteins that are prone to aggregate from supersaturated proteins that are not.

Upon examining the proteins that aggregate in U2OS cells, we found further support for this. Baseline aggregation in U2OS cells is also made up of supersaturated, LLPS-prone proteins (*Figure 3—figure supplement 1H–R*). Despite the baseline aggregation being far more pronounced in U2OS cells than in HEK293T cells, many supersaturated proteins are not SDS-insoluble in U2OS cells, even in cells treated with CPT or in cells lacking *ATM* (*Figure 3—figure supplement 1O*). In U2OS *ATM* KO cells, proteins that aggregate more are supersaturated compared to U2OS NIA proteins

(*Figure 3—figure supplement 1M*); for CPT-treated cells, this is not the case (*Figure 3—figure supplement 1M–O*). Proteins that aggregate more in U2OS *ATM* KO cells also have a higher general propensity to engage in LLPS as predicted by PScore and catGRANULE (*Figure 3—figure supplement 1P and Q*), while proteins that aggregate more in CPT-treated cells are enriched for proteins with a PScore > 4 (*Figure 3—figure supplement 1R*). From this, we conclude that both CPT treatment and a loss of ATM further exacerbate the aggregation of LLPS-prone and supersaturated proteins in a cell-type-dependent manner.

A GO term analysis of our RNAseq data revealed a striking lack of overlap in transcriptional processes altered upon treatment with CPT or loss of ATM function in either HEK293T or U2OS cells (*Figure 3—figure supplement 2*, *Supplementary file 2*). Intriguingly, only two transcripts (one up and one down) were significantly altered ($-1 > \log_2\text{FC} > 1$) after ATM inhibition in HEK293T cells despite the enhanced aggregation occurring in these cells. This further underlines that the enhanced aggregation after CPT treatment or a loss of ATM function is mostly driven by the physicochemical characteristics of the proteins involved.

## Genotoxic stress-induced protein aggregation is the result of a global lowering of the protein aggregation threshold

Our data shows that a substantial number of inherently similarly vulnerable proteins aggregate under the genotoxic conditions of CPT treatment or ATM loss. Their consistent aggregation across independent repeats argues against the possibility that this is caused by any genotoxic stress-induced DNA sequence alterations in their own coding regions as these would occur more randomly throughout the genome. Moreover, we find that the increased aggregation can also not be explained by any changes in abundance of the proteins involved, resulting for example from DNA damage-induced transcriptional dysregulation, as very limited overlap exists between proteins that aggregate and proteins with an altered expression upon CPT treatment or ATM loss (see *Figure 4A* for HEK293T and *Figure 4—figure supplement 1A* for U2OS).

Instead, our data indicate that a long-term consequence of these genotoxic conditions is a global lowering of the aggregation threshold of proteins. As a result, more and more LLPS-prone, supersaturated proteins that are normally largely soluble now start to aggregate, with the most vulnerable proteins aggregating first. This aggregation threshold appears to be inherently lower in U2OS cells compared to HEK293T cells, causing a large population of metastable proteins to aggregate already under normal conditions. Genotoxic stress in U2OS cells lowers the aggregation threshold even further, causing a 'second layer' of LLPS-prone proteins that are not even always supersaturated to aggregate also (*Figure 4—figure supplement 1B*).

This lowering of the aggregation threshold is highly reminiscent of 'classic' protein aggregation resulting from (chronic) proteotoxic stresses (*Weids et al., 2016*) and has been referred to as a disturbed (*Hipp et al., 2019*) or shifted protein homeostasis (*Ciryam et al., 2013*). In line with this, we find that proteins that aggregate more in HEK293T and U2OS cells after CPT treatment or after a (functional) loss of ATM are enriched for proteins that have been reported to aggregate upon heat treatment of cells (*Figure 4B*, *Figure 4—figure supplement 1C*; *Mymrikov et al., 2017*). In addition, they are enriched for constituents of stress granules (*Figure 4C*, *Figure 4—figure supplement 1D*; http://rnagranuledb.lunenfeld.ca), cellular condensates that have been found to function as nucleation sites for protein aggregation (*Dobra et al., 2018*; *Mateju et al., 2017*). Protein aggregation after heat shock and the formation of aberrant stress granules also includes the aggregation of newly synthesized proteins (*Ganassi et al., 2016*; *Xu et al., 2016*). To assess the extent to which newly synthesized proteins aggregate in cells exposed to genotoxic conditions, we pulsed HEK293T cells with $^{35}$S-labeled cysteine and methionine 48 hr after CPT treatment (*Figure 4—figure supplement 1E*). Notably, protein synthesis is reduced approximately threefold in CPT-treated cells (*Figure 4D*). Despite this strong reduction in protein synthesis, radioactively labeled proteins were still clearly present in the aggregating fraction of CPT-treated cells. They were however not enriched in the aggregating fraction compared to control cells (*Figure 4D*), indicating that the enhanced aggregation triggered by CPT is not explained through an accelerated aggregation of specifically newly synthesized proteins.

A shift in protein homeostasis has also been suggested to be key to the build-up of protein aggregates during aging (*Ciryam et al., 2014*) and to the initiation of protein aggregation in a range of chronic disorders (*David et al., 2010*; *Hipp et al., 2019*; *Morley et al., 2002*). Intriguingly, we find

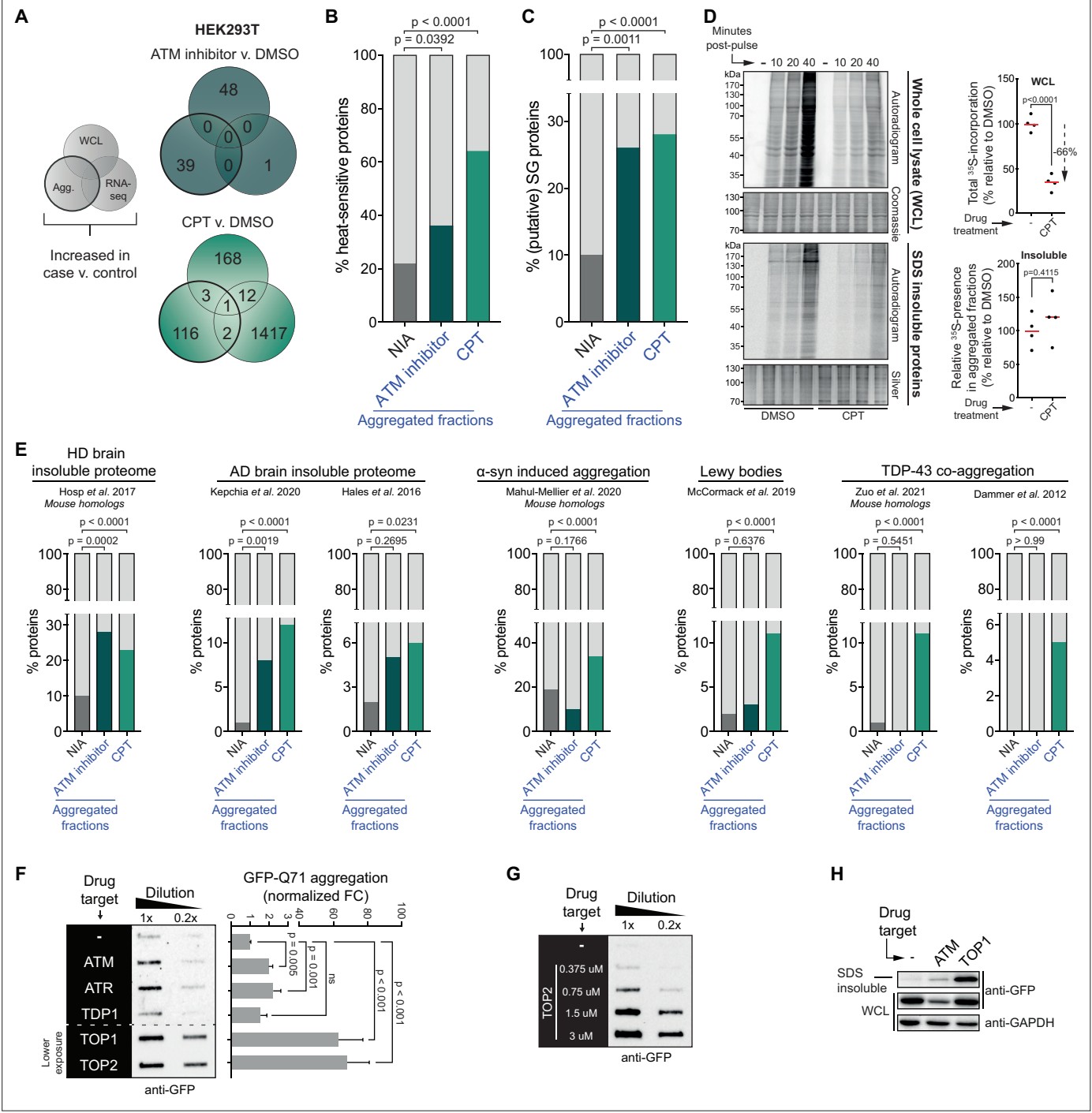

**Figure 4.** The cell-intrinsic aggregation threshold is lowered upon targeting ataxia telangiectasia mutated (ATM), ataxia telangiectasia and Rad3-related (ATR), or DNA topoisomerases. See also *Figure 4—figure supplements 1 and 2*. (**A**) Overlap between RNA-sequencing analysis and label-free quantification (LFQ) MS/MS analysis for whole-cell lysate (WCL) and aggregated (Agg.) protein fractions. Only significant increases are taken into account. (**B**) Relative occurrences in the indicated fractions in HEK293T cells of proteins that have been shown to aggregate upon heat stress. See text for reference. (**C**) Relative occurrences in the indicated fractions in HEK293T cells of proteins that have been found to be associated with stress granules. See text for reference. (**D**) Pulse label experiment. See also *Figure 4—figure supplement 1E*. HEK293T cells were treated with DMSO or camptothecin (CPT) (0,02 μM) and pulsed with radioactive 35S-labeled cysteine and methionine for 30 min. Cells were then harvested at the indicated timepoints (10, 20, or 40 min post pulse). Left upper panel: WCLs were run on SDS-PAGE and exposed (autoradiogram) or stained (Coomassie). Left lower panel: aggregated fractions were run on SDS-PAGE and exposed (autoradiogram) or stained (silver stain). Right upper panel: quantification of the incorporated 35S of the indicated treatment in the whole-cell extract. Right lower panel: quantification of the incorporated 35S of the indicated treatment in pellet

*Figure 4 continued on next page*

*Figure 4 continued*

fractions, normalized for total aggregation levels (as measured in the silver stains) and total $^{35}$S incorporation (as measured in the WCL fractions). n = 4. (**E**) Relative occurrences of proteins identified in various disease (model) datasets in the indicated fractions in HEK293T cells, obtained from the indicated studies. See also *Figure 4—figure supplement 2*. (**F**) Left panel: filter trap assay of HEK293 cells expressing inducible Q71-GFP that received the indicated treatment, probed with GFP antibody. n = 3. For doses, see *Table 1*. Right panel: quantification, using Student's two-tailed *t*-test followed by a Bonferroni correction for multiple comparisons. See also *Figure 4—figure supplement 1F*. (**G**) Filter trap assay of HEK293 cells expressing inducible Q71-GFP that were treated with the indicated doses of etoposide (Etop), probed with GFP antibody. n = 2. (**H**) Western blot of WCL and aggregated proteins isolated from HEK293 cells expressing inducible luciferase-GFP, treated with ATM inhibitor or CPT, probed with the indicated antibodies. n = 2. See also *Figure 4—figure supplement 1K*. In (**B**), (**C**), and (**E**), chi-square testing was used to evaluate the statistical significance of differences in distributions. In (**D**), two-tailed Student's *t*-tests were used.

The online version of this article includes the following source data and figure supplement(s) for figure 4:

**Source data 1.** Data from *Figure 4D*.

**Source data 2.** Data from *Figure 4F*.

**Source data 3.** Data from *Figure 4G*.

**Source data 4.** Data from *Figure 4H*.

**Figure supplement 1.** Increased aggregation triggered by camptothecin treatment and ataxia telangiectasia mutated (ATM) loss overlaps with that occurring in various proteinopathies.

**Figure supplement 1—source data 1.** Data from *Figure 4—figure supplement 1F*.

**Figure supplement 1—source data 2.** Data from *Figure 4—figure supplement 1G*.

**Figure supplement 1—source data 3.** Data from *Figure 4—figure supplement 1H*.

**Figure supplement 2.** Proteins that aggregate in HEK293T treated with camptothecin (CPT) are linked to various proteinopathies.

that proteins that aggregate after transient CPT treatment are enriched for constituents of various disease-associated protein aggregates (*Figure 4E*). 67% (82/122) of them – or their mouse homologs – have already previously been identified in TDP-43 aggregates (*Dammer et al., 2012*; *Zuo et al., 2021*), Lewy bodies (*McCormack et al., 2019*), or α-synuclein-induced aggregates (*Mahul-Mellier et al., 2020*), or found to aggregate in Huntington's disease (HD) (*Hosp et al., 2017*) or Alzheimer's disease (AD) brains (*Hales et al., 2016*; *Kepchia et al., 2020 Figure 4—figure supplement 2*). An enrichment for HD and AD brain aggregating proteins was also observed among proteins that aggregate after inhibition of ATM (*Figure 4E*).

If genotoxic conditions indeed over time lead to a lowering of the aggregating threshold, this would predict that they can also result in an accelerated aggregation of aggregation-prone model substrates. For example, disease-associated expanded polyQ proteins are inherently aggregation prone, and they have been shown to aggregate faster in systems in which protein homeostasis is impaired (*Gidalevitz et al., 2013*; *Gidalevitz et al., 2010*). We went back to HEK293 cells and employed a line carrying a stably integrated, tetracycline-inducible GFP-tagged Huntingtin exon 1 containing a 71 CAG-repeat (encoding Q71). Transient targeting of ATM, ATR, and in particular TOPs, but not TDP1, 24–48 hr prior to the expression of polyQ (*Figure 4—figure supplement 1F*) indeed accelerated polyQ aggregation in these cells (*Figure 4F*, *Figure 4—figure supplement 1G*), closely mirroring the increased aggregation that we observed before (*Figure 1C and D*). The accelerated polyQ aggregation under these conditions is also dose-dependent (*Figure 4G*, *Figure 4—figure supplement 1H and I*), and it is not explained by changes in total polyQ levels (*Figure 4—figure supplement 1G–I*). PolyQ aggregation is normally proportional to the length of the CAG repeat, which is intrinsically unstable. Importantly, we find no evidence that the accelerated polyQ aggregation induced by these genotoxic conditions can be explained by an exacerbated repeat instability (*Figure 4—figure supplement 1J*). Next, we also used the same tetracycline-inducible system and experimental set-up to investigate the aggregation of the protein folding model substrate luciferase-GFP (*Figure 4—figure supplement 1K*). We find that transient targeting of either ATM or TOP1 results in an enrichment of luciferase-GFP in the aggregated fraction (*Figure 4H*).

## Genotoxic stress results in a rewiring of chaperone networks, which is however insufficient to prevent client aggregation

We noted that ATM inhibition and in particular CPT treatment resulted in an increased aggregation of multiple (co)chaperones in HEK293T cells (*Figure 5—figure supplement 1A and B*). In U2OS

cells, many (co)chaperones are already aggregating regardless of exposure to genotoxic stress, but still several chaperones aggregated significantly more in *ATM* KO cells or in cells treated with CPT (*Figure 5—figure supplement 1C and D*). The overlap that exists between aggregating chaperones in each cell line suggests that this occurs mostly cell line specific (*Figure 5A*, *Figure 5—figure supplement 1E*). These findings are interesting as chaperone systems have the ability to modulate aggregation (*Hartl et al., 2011*; *Mogk et al., 2018*; *Sinnige et al., 2020*; *Tam et al., 2006*). HSP70s (HSPAs) are among the most ubiquitous chaperones, and they have been shown to play a key role in maintaining protein homeostasis in virtually all domains of life (*Gupta and Singh, 1994*; *Hunt and Morimoto, 1985*; *Lindquist and Craig, 1988*). Upon cross-referencing the NIA and aggregating fractions against a recently generated client database of HSPA8 (HSC70; constitutively active form of HSP70) and HSPA1A (constitutively active and stress-inducible HSP70) (*Ryu et al., 2020*), we find that HSPA8 and HSPA1A clients are enriched among aggregating proteins (*Figure 5B*). We also mined the BioGRID human protein-protein interaction database using the complete KEGG dataset of (co) chaperones (168 entries). Although the transient and energetically weak nature of the interactions between many (co)chaperones and their clients (*Clouser et al., 2019*; *Kampinga and Craig, 2010*; *Mayer, 2018*) makes it likely that these interactions are underrepresented in the BioGRID database, it can provide additional insight into the presence of (putative) chaperone clients in the aggregating fractions (*Victor et al., 2020*). We find that all aggregating fractions are enriched for (co)chaperone interactors compared to nonaggregating proteins (NIA) (*Figure 5C*, *Figure 5—figure supplement 2*). Aggregating proteins have reported interactions with a broad range of chaperone families, most notably HSP70s and HSP90s (and known co-factors of these), and chaperonins (TRiC/CCT subunits) (*Figure 5D*, *Figure 5—figure supplement 2*).

Intriguingly, several (co)chaperones that we found to aggregate themselves are among the most frequent interactors (*Figure 5D*). This suggests that they were sequestered by protein aggregates as they engaged their client proteins, in line with what has been reported for disease-associated aggregation (*Hipp et al., 2019*; *Jana et al., 2000*; *Kim et al., 2013*; *Mogk et al., 2018*; *Yu et al., 2019*; *Yue et al., 2021*). Overall, we find that the relative levels of chaperone engagement of the different aggregating fractions largely reflect their respective supersaturation and LLPS propensities.

When the capacity of chaperone systems is overloaded, this can eventually trigger a rewiring of chaperone systems. This plasticity allows cells to adapt to varying circumstances and proteotoxic stress conditions (*Klaips et al., 2018*). In HEK293T cells, we find that treatment with CPT results in an overall upward shift of (co)chaperone expression levels, as measured in both our WCL MS/MS analysis (16 up, 8 down) (*Figure 5E*) and in our RNAseq dataset (11 up, 5 down) (*Figure 5F*). Upregulated chaperones include HSPB1, DNAJA1, HSPA5, HSPA8, and HSP90AA1 (*Figure 5E*), all of which are among the most frequent interactors of aggregating proteins in CPT-treated cells. The expression of most of these chaperones is regulated by the heat shock factor 1 (HSF1) transcription factor (*Metchat et al., 2009*; *Neueder et al., 2017*; *Ostling et al., 2007*; *Trinklein et al., 2004*). HSF1 is indeed partially activated by CPT treatment in HEK293T cells (*Figure 5G*). ATM inhibition in HEK293T cells resulted in a marginal HSF1 activation. This is in line with an overall less pronounced aggregation response after ATM inhibition in HEK293T cells, which appears to be insufficient to initiate a clear rewiring of chaperone systems (*Figure 5—figure supplement 1F*, *Supplementary file 2*). In U2OS cells, a loss of ATM or treatment with CPT appears to result in a more balanced rewiring of chaperone systems (*Figure 5—figure supplement 1G–J*). In CPT-treated U2OS cells, protein levels of multiple chaperones are even lowered. Nevertheless, similar to HEK293T cells, many of the most frequent (co) chaperone interactors of the aggregating proteins in U2OS are found to aggregate themselves as well. These findings indicate that (sufficient) genotoxic stress induces a rewiring of chaperone systems in a cell and stress-specific manner. This rewiring is however insufficient to prevent the increased aggregation of metastable client proteins.

We reasoned that the difference in aggregation between HEK293T and U2OS cells might also be reflected in different chaperone expression levels already under normal conditions. Indeed, a differential expression analysis between untreated HEK293T and untreated U2OS cells revealed a strong overall upward shift of (co)chaperone gene expression levels in the latter (*Figure 5—figure supplement 1K*). For example, we found that gene expression levels of the small heat shock-like protein Clusterin (CLU) are >100-fold higher in wild-type U2OS compared to HEK293T cells, gene expression levels of the stress-inducible HSPA1A are >150-fold higher, and that of HSPB5 (or CRYAB, i.e.,

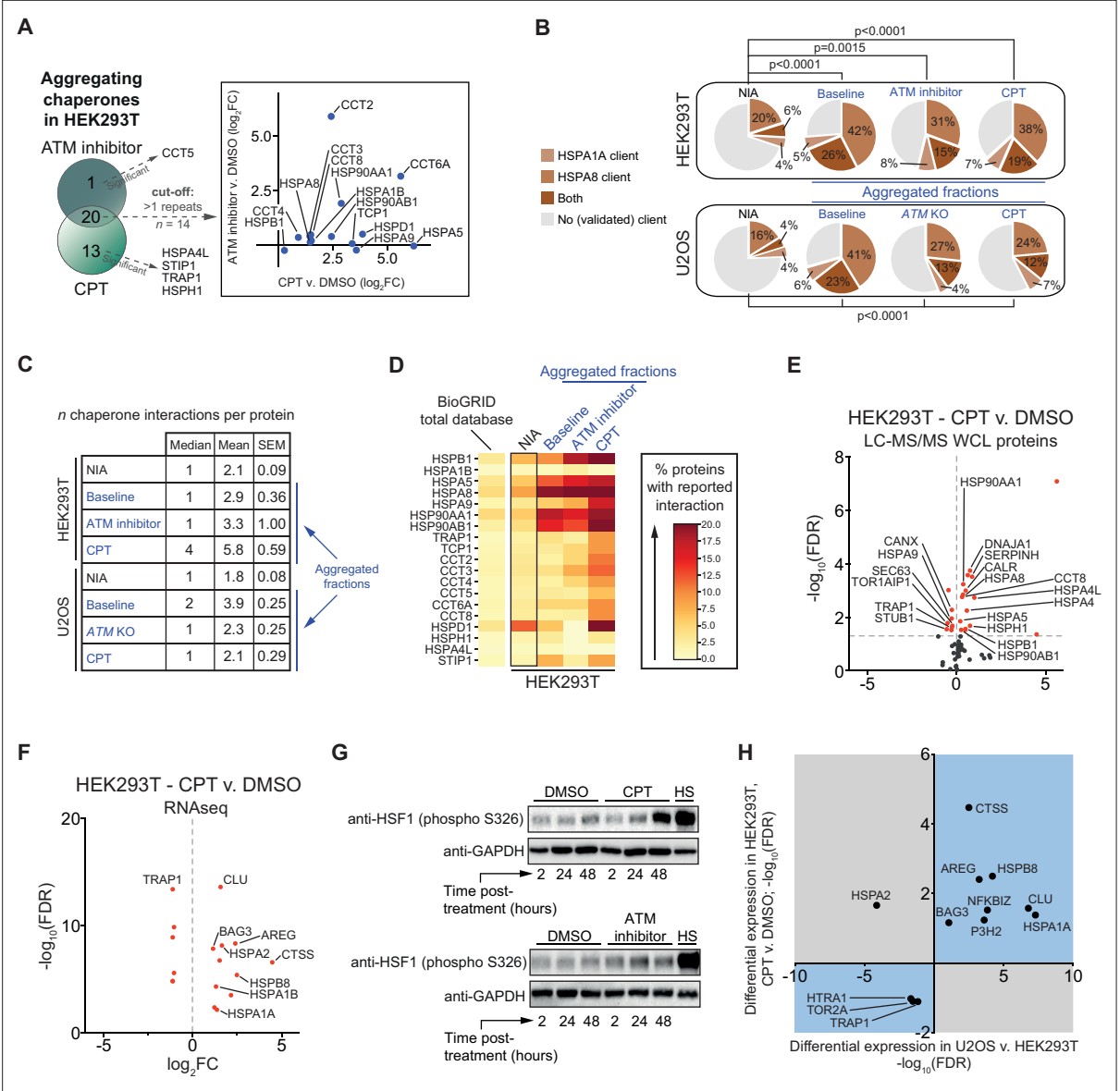

**Figure 5.** The lowered aggregation threshold caused by topoisomerase poisoning or a loss of ataxia telangiectasia mutated (ATM) is accompanied by a rewiring and aggregation of known interacting (co)chaperones. See also *Figure 5—figure supplements 1 and 2*. (**A**) Presence of (co)chaperones in the aggregated protein fractions in HEK293T cells. Left panel: Venn diagram showing the overlap in aggregating chaperones. Right panel: graph depicting the overlap in aggregating chaperones in detail. (**B**) Pie charts showing the presence of HSPA1A and HSPA8 clients in aggregated protein fractions compared to clients present in both NIA fractions. See text for reference; only clients identified in at least two out of three repeats in Lee et al. were taken into account here. (**C**) Table showing the number of (co)chaperones logged in BioGRID as interacting with NIA and aggregating protein fractions. (**D**) See also (**A**): only (co)chaperones aggregating in both HEK293T experimental set-ups or significantly more in one of them are shown here. BioGRID (co)chaperone interactions with the aggregated proteins identified in this study, per (co)chaperone. Darker colors represent a higher percentage of proteins with a reported binding to that (co)chaperone. See *Figure 5—figure supplement 2* for a complete overview. (**E**) Volcano plot showing protein levels of (co)chaperones in camptothecin (CPT)-treated HEK293T cells compared to DMSO-treated cells. (**F**) Differentially expressed (co)chaperones in CPT-treated HEK293T cells compared to DMSO-treated cells based on RNAseq data. (**G**) Western blot analysis using the indicated antibodies on the whole-cell lysate (WCL) fractions of HEK293T cells after the indicated time and treatment. HS, heat shock (2 hr at 43°C incubator). n = 2. (**H**) Graph showing (co)chaperones whose genes show a differential expression in both CPT-treated HEK293T cells compared to DMSO-treated HEK293T cells and in untreated U2OS compared to untreated HEK293T cells.

The online version of this article includes the following source data and figure supplement(s) for figure 5:

**Source data 1.** Data from *Figure 5G*.

*Figure 5 continued on next page*

*Figure 5 continued*

**Figure supplement 1.** Chaperone systems are rewired in line with the presence of chaperone clients in aggregates induced by camptothecin or ataxia telangiectasia mutated (ATM) loss.

**Figure supplement 2.** Heatmaps of chaperone interactions of aggregating fractions.

αB-crystallin) were >400-fold higher. Interestingly, the differences in expression of chaperone systems in U2OS compared to HEK293T overlap with the changes occurring after CPT treatment in the latter. Out of the 13 (co)chaperones identified to be expressed differently in both (RNAseq; –1 > log2FC > 1), 12 are altered in the same direction (*Figure 5H*).

## Genotoxic stress-induced protein aggregation is amenable to modulation by chaperone systems

Our data suggest that the lowering of the aggregation threshold upon various genotoxic conditions is caused by an overload of chaperone systems, leading to a shift in protein homeostasis. We reasoned that targeting chaperone systems may then exacerbate aggregation. Indeed, mild HSP70 inhibition using the HSP70/HSC70 inhibitor VER-155008 after CPT treatment increased CPT-induced protein aggregation even further, while having no clear impact on aggregation in control cells (*Figure 6— figure supplement 1A*). Similar results were obtained when we blotted the aggregated fractions for MCM7 and TUBA1A (*Figure 6A*).

We next reasoned that increasing chaperone capacity may also raise the aggregation threshold again. We screened an overexpression library of several major chaperone families, including HSPAs, J-domain proteins (JDPs), and small heat shock proteins (HSPBs) for their ability to reduce the increased protein aggregation triggered by genotoxic conditions using U2OS *ATM* KO cells as a model (*Figure 6—figure supplement 1B*). While most of these had no overt effect, overexpression of several JDPs reduced protein aggregation, including the generic anti-amyloidogenic protein DNAJB6b (*Aprile et al., 2017*; *Hageman et al., 2010*). However, we found that the small heat shock protein HSPB5 was especially effective. HSPB5 is a potent suppressor of aggregation and amyloid formation (*Delbecq and Klevit, 2019*; *Golenhofen and Bartelt-Kirbach, 2016*; *Hatters et al., 2001*; *Webster et al., 2019*). Its higher expression in U2OS cells compared to HEK293T cells, as well as its further upregulation in U2OS cells lacking *ATM*, suggests that it plays an important role in counteracting widespread protein aggregation in these cells.

We generated U2OS cells that stably overexpress HSPB5 in both wild-type and *ATM*-deficient backgrounds (*Figure 6—figure supplement 1C*), and confirmed that this drastically reduced the enhanced protein aggregation in the latter (*Figure 6B and C*). HSPB5 overexpression also reduced ProteoStat aggresome staining and the occurrence of cytoplasmic FUS puncta (without affecting overall FUS levels; see *Supplementary file 1*), two other markers of a disrupted protein homeostasis (*Figure 6D–G*; *Neumann et al., 2006*; *Shen et al., 2011*).

Although HSPB5 itself has never been linked to genome maintenance, we evaluated whether HSPB5 can mitigate the increased aggregation following a loss of ATM in U2OS cells by altering DNA repair capacity. However, we found no indication for this as the gamma irradiation-induced DNA lesion accumulation and subsequent resolution as measured by 53BP1 foci formation was not affected by HSPB5 expression in neither U2OS wild-type nor *ATM* KO cells (*Figure 6—figure supplement 1D and E*). Moreover, neither HSPB5, nor HSP70 nor HSP90 accumulated at either CPT- or gamma irradiation-induced DNA damage sites (*Figure 6—figure supplement 2A–C*). This points out that the overload of chaperones is not due to a sequestering to DNA damage sites.

HSPB5 is one of two chaperones that are transcriptionally upregulated in U2OS cells after either a loss of ATM or CPT treatment, and the only one that is not transcriptionally upregulated in CPT-treated HEK293T cells (*Figure 6—figure supplement 1F*). Crucially, we found that overexpression of HSPB5 can reduce the enhanced protein aggregation in CPT-treated U2OS cells as well (*Figure 6H, I*), but that stable overexpression of HSPB5 has no effect on the CPT-induced aggregation in HEK293 cells (*Figure 6—figure supplement 1G*). These data emphasize that the rewiring of chaperone systems in response to genotoxic stress is tailored to each cell, depending largely on the ground state of protein homeostasis and concomitant aggregation that occurs.

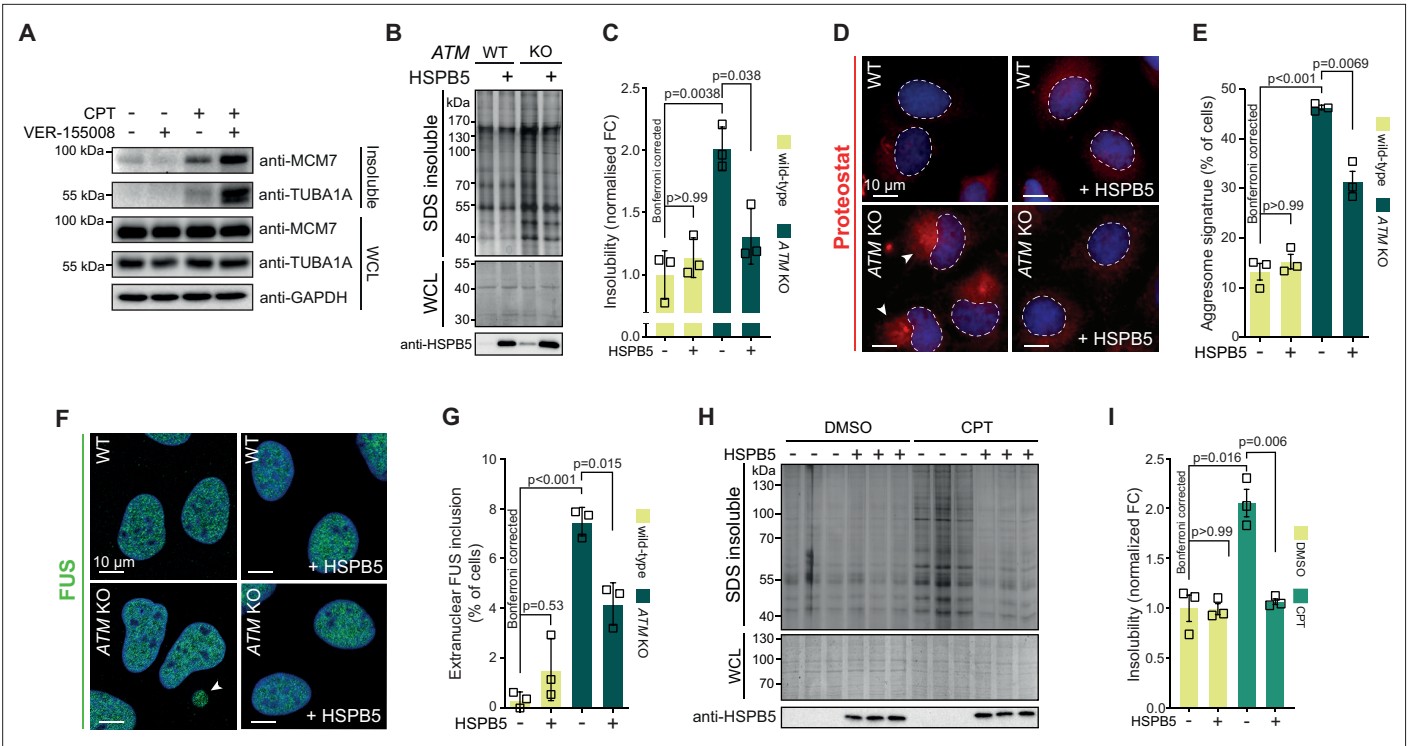

**Figure 6.** Protein aggregation triggered by genotoxic stress is amenable to modulation by chaperones. See also *Figure 6—figure supplements 1 and 2*. (**A**) Western blot of whole-cell lysate (WCL) and aggregated proteins isolated from HEK293T cells treated transiently with DMSO or camptothecin (CPT), followed by treatment with the VER-155008 HSP70 inhibitor (10 μM), probed with the indicated antibodies. n = 3. (**B**) Aggregated (silver stain) and WCL (Coomassie) fractions of U2OS wild-type and *ATM* KO cells, with or without overexpression of HSPB5. (**C**) Quantification of (**B**). (**D**) Representative immunofluorescence pictures of U2OS wild-type and *ATM* KO cells stably overexpressing HSPB5 or not, stained with ProteoStat (red) and Hoechst (blue). (**E**) Quantification of aggresome signatures in (**D**). (**F**). Representative immunofluorescence pictures of U2OS wild-type and *ATM* KO cells stably overexpressing HSPB5 or not, stained with anti-FUS (green) and Hoechst (blue). (**G**) Quantification of extranuclear FUS inclusions in (**F**). (**H**) Aggregated (silver stain) and WCL (Coomassie) fractions of HEK293 cells stably overexpressing HSPB5 or not, treated transiently with DMSO or CPT. Three technical repeats are shown here. (**I**) Quantification of three independent biological repeats of (**H**). In (**C**), (**E**), (**G**), and (**I**), squares represent independent biological repeats, bars represent mean ± SEM. p-Values are obtained by two-tailed Student's *t*-tests followed by a Bonferroni correction for multiple comparisons.

The online version of this article includes the following source data and figure supplement(s) for figure 6:

**Source data 1.** Data from *Figure 6A*.

**Source data 2.** Data from *Figure 6B*.

**Source data 3.** Data from *Figure 6H*.

**Figure supplement 1.** HSPB5 alleviates protein aggregation triggered by a loss of ataxia telangiectasia mutated (ATM) in U2OS cells independent of overt DNA repair capacity changes.

**Figure supplement 1—source data 1.** Data from *Figure 6—figure supplement 1A*.

**Figure supplement 1—source data 2.** Data from *Figure 6—figure supplement 1B*.

**Figure supplement 1—source data 3.** Data from *Figure 6—figure supplement 1C*.

**Figure supplement 1—source data 4.** Data from *Figure 6—figure supplement 1G*.

**Figure supplement 2.** HSPB5, HSP70, and HSP90 do not (re-)localize to DNA damage sites.

## Discussion

Here, we report that TOP poisoning and functional impairment of ATM or ATR trigger a widespread aggregation of LLPS-prone and supersaturated proteins. Our data show that the aggregation of these metastable proteins is a consequence of an overload of chaperone systems under these genotoxic conditions. This is illustrated by the aggregation of certain chaperones, and the observation that specifically the (putative) clients of these chaperones aggregate as well. It is further supported by the notion that CPT treatment leads to a strong reduction in protein synthesis over time, something that has been reported for other forms of DNA damage as well (*Halim et al., 2018*). Reduced protein

synthesis is a well-known response to proteotoxic stress and is believed to lower the strong demand for chaperone capacity of nascent chains that are innately vulnerable to misfolding and aggregation (*Balchin et al., 2016*). This indicates that despite a reduction in protein synthesis, genotoxic stress still leads to an overload of chaperone systems, causing protein homeostasis to shift. This effectively lowers the cell-intrinsic threshold of protein aggregation, and as a result, vulnerable proteins that are largely kept soluble under normal conditions now succumb more readily to aggregation. The accelerated aggregation of the model substrates polyQ- and luciferase that occurs in cells exposed to these conditions underlines this threshold change as well.

The observed shift in protein homeostasis after genotoxic stress is strikingly reminiscent of what is believed to occur under conditions of (chronic) stress (*Weids et al., 2016*) and during many age-related neurodegenerative disorders (*David et al., 2010*; *Hipp et al., 2019*; *Morley et al., 2002*). Supersaturated proteins have been found to be overrepresented in cellular pathways associated with these disorders (*Ciryam et al., 2015*), and disease-associated aggregating proteins, including FUS, tau, and α-synuclein, are known to exhibit LLPS behavior (reviewed in *Zbinden et al., 2020*). Indeed, we find that the proteins that aggregate in our experiments show a strong overlap in identity and function with stress-induced aggregation, and with the aggregation observed in various proteinopathies.

The shift in protein homeostasis under genotoxic stress conditions can theoretically be caused by either an altered capacity of protein quality control systems or by an increased demand emanating from an altered proteome. These are, however, difficult to disentangle fully, in particular because they may form a vicious cycle of events, where (co)chaperones are increasingly sequestered as a growing number of proteins succumbs to aggregation (*Klaips et al., 2018*). Either way, both result in a net lack of protein quality control capacity, which can be rescued by upregulating specific chaperones, and exacerbated by further decreasing chaperone capacity. The aggregation that occurs under the genotoxic conditions used in our study follows this pattern. Nevertheless, our data lead us to hypothesize that the overload of chaperone systems is largely caused by an increased demand for chaperone activity in the proteome. Multiple (co)chaperones that have been reported to interact frequently with the aggregating proteins are upregulated under the genotoxic conditions used in our study. Many of these (co)chaperones aggregate themselves as well. Crucially, further overexpression of one of the most upregulated chaperones in U2OS, HSPB5, is able to largely bring aggregation in CPT-treated and *ATM* KO cells back down to the control level.

The strong upregulation of several small heat shock (-like) proteins in U2OS cells, including HSPB5, seems to point at a rewiring of the chaperone network in this cell line towards a more prominent reliance on this class of chaperones. Small heat shock proteins have been reported to act together in heterodimers and hetero-oligomers (*Aquilina et al., 2013*; *Mymrikov et al., 2020*). In addition, in vivo they rely on other chaperone systems such as the HSP70 machinery to efficiently counteract aggregation (*Mogk et al., 2003*; *Reinle et al., 2022*; *Zwirowski et al., 2017*). The low expression of small heat shock proteins in general and of HSPB5 specifically in HEK293(T) cells suggests that these cells might not be equipped to wield elevated levels of HSPB5 to prevent widespread aggregation of proteins. This may explain why elevated levels of HSPB5 have no effect on CPT-induced aggregation in HEK293 cells.

The strong overlap between CPT-induced aggregation in HEK293T cells and baseline aggregation in U2OS cells also argues for an increased demand. U2OS is a cancer cell line (osteosarcoma), whereas HEK293(T) cells have a vastly different origin (embryonic kidney). Cancer cells inherently exhibit elevated levels of protein stress, which has been attributed to an increased protein folding and degradation demand (*Dai et al., 2012*; *Deshaies, 2014*). The notion that the rewiring of chaperone systems in response to CPT treatment in HEK293T cells mimics the difference in chaperone wiring between HEK293T and U2OS cells underlines this further.

Our data indicate that any increased demand caused by these genotoxic conditions is, however independent of quantitative changes of the aggregating proteins themselves and likely also of any genetic alterations in their coding regions (in *cis* genetic alterations). The accelerated polyQ aggregation – not accompanied by any enhanced CAG repeat instability – provides support for this. This is not necessarily surprising as proteins that aggregate as a consequence of an overload of the protein quality control do not have to be altered themselves. Previous studies have shown that during proteomic stress a destabilization of the background proteome can result in a competition for the limited chaperone capacity available, causing proteins that are highly dependent on

chaperones for their stability and solubility to aggregate readily (*Gidalevitz et al., 2010*; *Gidalevitz et al., 2011*).

In this light, it is interesting that the proteins that aggregate in our experiments are in general prone to engage in LLPS. LLPS is known to be regulated by RNA and often involves RNA-binding proteins. Strikingly, we find that the aggregation that occurs after genotoxic stress is enriched for RNA-binding proteins. This enhanced aggregation can also be mitigated by inhibiting PARylation, which plays a key role in the regulation of LLPS processes (*Duan et al., 2019*; *McGurk et al., 2018*). LLPS is a different biochemical process than protein aggregation (with different underlying mechanisms and principles), but aberrant LLPS can drive the nucleation of insoluble (fibrillar) protein aggregates, for example, for polyQ (*Peskett et al., 2018*). It is therefore believed that LLPS events need to be closely regulated and monitored to prevent aberrant progression into a solid-like state (*Alberti and Dormann, 2019*). Although data is so far limited, chaperones, and in particular small heat shock proteins, have been reported to play a pivotal role in the surveillance of biomolecular condensates. For example, the HSPB8-BAG3-HSPA1A complex has been found to be important for maintaining stress granule dynamics (*Ganassi et al., 2016*), and recent work uncovered that HSPB1 is important to prevent aberrant phase transitions of FUS (*Liu et al., 2020*). We find that HSPB8, BAG3, and HSPA1A are upregulated in HEK293T cells treated with CPT. Interestingly, although HSPB5 itself has so far not been shown to undergo LLPS, like HSPB1, it has been found to associate with nuclear speckles (*van den IJssel et al., 1998*), which are membraneless as well. HSPB5 has also been shown to be important to maintain the stability of the cytoskeleton (*Ghosh et al., 2007*; *Golenhofen et al., 1999*; *Yin et al., 2019*), and we find that many proteins that aggregate upon genotoxic stress conditions are cytoskeleton (-related) components. A growing body of evidence indicates that cytoskeleton organization is regulated through LLPS processes (reviewed in *Wiegand and Hyman, 2020*). Our data thus indicate that genotoxic stress conditions can exacerbate the risk of aberrant progression of LLPS processes, which in turn may trigger an overload of chaperone systems.

The increased protein aggregation that occurs after a loss of ATM – including in A-T patient brains – has been recently attributed to an accumulation of DNA damage (*Lee et al., 2021*). As an impaired response to DNA damage is believed to be the primary driving force of A-T phenotypes (*Shiloh, 2020*), these findings have fueled the idea that a disruption of protein homeostasis may be an important disease mechanism in A-T. Our data provide further support for this as they show that the widespread aggregation caused by a loss of ATM follows a predictable pattern that overlaps strikingly with the aggregation that is believed to underlie many neurodegenerative disorders. Importantly, our findings also provide a proof of principle that other genotoxic conditions – including chemotherapeutic TOP poisons – can have a very similar impact. This points at the existence of a broader link between DNA damage and a loss of protein homeostasis . Although further research is needed to determine the full breadth and relevance of this link, our work may thus offer clues as to why besides impairments in ATM many other genome maintenance defects are characterized by often overlapping (neuro)degenerative phenotypes as well (*Petr et al., 2020*).

# Materials and methods

**Key resources table**

| Reagent type (species) or resource | Designation | Source or reference | Identifiers | Additional information |
|---|---|---|---|---|
| Cell line (human) | HEK293T | ATCC | CRL-3216 | |
| Cell line (human) | HEK293 | ATCC | CRL-1573 | |
| Cell line (human) | HEK293 *ATM* KO | This study | | See 'Mammalian cell culture' |
| Cell line (human) | HEK293 HTT Q71-GFP | PMID:20159555 | | See 'Mammalian cell culture' |
| Cell line (human) | HEK293 luciferase-GFP | PMID:21231916 | | See 'Mammalian cell culture' |
| Cell line (human) | HEK293 + HSPB5 | This study | | See 'Mammalian cell culture' |

*Continued on next page*

*Continued*

| Reagent type (species) or resource | Designation | Source or reference | Identifiers | Additional information |
|---|---|---|---|---|
| Cell line (human) | U2OS | ATCC | HTB-96 | |
| Cell line (human) | U2OS *ATM* KO | This study | | See 'Mammalian cell culture' |
| Cell line (human) | U2OS + HSPB5 | This study | | See 'Mammalian cell culture' |
| Cell line (human) | U2OS *ATM* KO + HSPB5 | This study | | See 'Mammalian cell culture' |
| Cell line (human) | Phoenix-Ampho | ATCC | RRID:CVCL_H716 | Retrovirus packaging cell line |
| Antibody | GFP (mouse, monoclonal) | Takara Bio Clontech | 632380 | WB (1:5000) |
| Antibody | ATM (mouse, monoclonal) | Santa Cruz | Sc-23921 | WB (1:200) |
| Antibody | HSPB5 (mouse, monoclonal) | StressMarq | SMC-159 | WB (1:2000) |
| Antibody | HSPB5 (mouse, monoclonal) | StressMarq | SMC-165 | IF (1:200) |
| Antibody | GAPDH (mouse, monoclonal) | Fitzgerald | 10R-G109a | WB (1:10,000) |
| Antibody | TUB (mouse, monoclonal) | Sigma-Aldrich | T5138 | WB (1:4000) |
| Antibody | HDAC1 (mouse, monoclonal) | DSHB | PCPR-HDAC1-2E12 | WB (0.5 µg/ml) |
| Antibody | MCM7 (Mmouse, monoclonal) | Santa Cruz | 47DC141 | WB (1:100) |
| Antibody | TUBA1A (mouse, monoclonal) | Sigma-Aldrich | T5168 | WB (1:2000) |
| Antibody | FUS (mouse, monoclonal) | Santa Cruz | Sc-47711 | IF (1:200) |
| Antibody | 53BP1 (rabbit, monoclonal) | Santa Cruz | Sc-22760 | IF (1:150) |
| Antibody | 53BP1 (rabbit, monoclonal) | Bethyl | A300-272A | IF (1:500) |
| Antibody | HSP70 (mouse, monoclonal) | StressMarq | SMC-104A | IF (1:100) |
| Antibody | HSP90 (mouse, monoclonal) | StressMarq | SMC-149 | IF (1:100) |
| Recombinant DNA reagent | pQCXIN–HSPB5 (plasmid) | PMID:20843828 | | |
| Recombinant DNA reagent | ATM CRISPR/Cas9 KO (plasmid) | Santa Cruz | sc-400192 | |
| Recombinant DNA reagent | ATM HDR (plasmid) | Santa Cruz | sc-400192-HDR | |
| Sequence-based reagent | HEK293Q71F (forward primer) | This study | PCR primer | GAGTCC CTCAAG TCCTTCC |
| Sequence-based reagent | HEK293Q71R (reverse primer) | This study | PCR primer | AAACGG GCCCTC TAGACTC |
| Commercial assay or kit | Silver stain kit | Pierce (Thermo Scientific) | 24612 | |
| Commercial assay or kit | Allprep DNA/RNA isolation mini kit | QIAGEN | 80004 | |
| Commercial assay or kit | S-trap micro | Protifi | K02-micro-10 | |
| Commercial assay or kit | Masterpure Complete DNA and RNA purificiation kit | Epicentre (supplied through Lucigen) | MC85200 | |
| Commercial assay or kit | QuantSeq 3' mRNA-Seq library prep kit (FWD) | Lexogen | 015.96 | |
| Chemical compound, drug | Camptothecin | Selleckchem | S1288 | See *Table 1* |
| Chemical compound, drug | Etoposide | Sigma-Aldrich | E1383 | See *Table 1* |
| Chemical compound, drug | TDP1 inhibitor | Merck | 532177 | See *Table 1* |

*Continued*

| Reagent type (species) or resource | Designation | Source or reference | Identifiers | Additional information |
|---|---|---|---|---|
| Chemical compound, drug | KU-55933 (ATM inhibitor) | Selleckchem | S1092 | See *Table 1* |
| Chemical compound, drug | KU 0058948 (PARP inhibitor) | Axon Medchem | 2001 | See *Table 1* |
| Chemical compound, drug | VE-821 | Axon Medchem | 1893 | See *Table 1* |
| Chemical compound, drug | VER-155008 | Axon Medchem | 1608 | (10 µM) |
| Chemical compound, drug | [S35]Met/cys | Hartmann Analytic | IS-103 | (10 µCi/ml) |
| Chemical compound, drug | ProteoStat | Enzo Life Sciences | ENZ-51023-KP050 | |
| Chemical compound, drug | AmpliTaq Gold Fast PCR mix | Applied Biosystems (supplied through Thermo Fisher) | 4390937 | |
| Software, algorithm | Prism | GraphPad | | |
| Software, algorithm | Illustrator 2021 | Adobe | | |
| Software, algorithm | TANGO | PMID:15361882 | | |
| Software, algorithm | CamSol Intrinsic | PMID:25451785 | | |
| Software, algorithm | catGRANULE | PMID:23222640 | | |
| Software, algorithm | PScore | PMID:29424691 | | |
| Software, algorithm | MaxQuant | PMID:19029910 | | |
| Software, algorithm | Lexogen QuantSeq 2.3.1 FWD UMI | BlueBee genomics (Illumina) | | |
| Software, algorithm | edgeR | PMID:19910308 | | |
| Software, algorithm | Cytoscape (in Python) | PMID:31477170 | | |
| Software, algorithm | Metascape (webserver) | PMID:30944313 | | |
| Software, algorithm | ImageJ (Fiji) | PMID:22930824 | | |
| Other | DMEM without methionine/cysteine | Gibco (supplied through Thermo Fisher) | 21013024 | |

## Statistical analyses

Statistical testing was performed using GraphPad Prism software, except for label-free quantification (LFQ) proteomics and RNA sequencing, which were analyzed in R (see their respective sections for more information). The statistical tests that were used are indicated in each figure legend. For experiments with pairwise comparisons, two-tailed Student's unpaired *t*-test was used unless otherwise indicated. For experiments with multiple comparisons, a Kruskal–Wallis with Dunn's post-hoc test (when datasets did not pass normality testing) or two-tailed Student's unpaired *t*-tests with Bonferroni correction (when indicated) was performed. p-Values are shown for all experiments. All repetitions (n) originate from independent replicates; any representation of technical repeats in figures is explicitly mentioned in the accompanying legend. Gels and stains were processed and analyzed using ImageJ software (Fiji).

## Mammalian cell culture

All parental cell lines were obtained from ATCC (see Key resources table) and are mycoplasma negative (GATC Biotech GA, Konstanz, Germany). Cell lines were cultured in DMEM (Gibco) supplemented with 10% FBS (Sigma-Aldrich), 100 units/ml penicillin, and 100 µg/ml streptomycin (Invitrogen). HEK293 cells expressing inducible GFP-Htt^exon1-Q71 (GFP-Q71) have been described previously (*Hageman et al., 2010*), and HEK293 cells expressing inducible luciferase-GFP as well (*Hageman et al., 2011*). U2OS and HEK293 *ATM* KO cells were generated using the ATM CRISPR/Cas9 KO and ATM HDR plasmids (sc-400192, sc-400192-HDR from Santa Cruz) according to the manufacturer's guidelines. Individual clones were picked and verified by PCR and Western blotting. For the generation of U2OS and HEK293 cells overexpressing HSBP5, see later section.

**Table 1.** Genotoxic drugs used in this study.

| Drug | Target | Concentration |
|---|---|---|
| Camptothecin | TOP1 | 20–600 nM. *Figure 1C*: 100 nM; *Figure 1E*: 20–100 nM; *Figure 1G*: 40 nM; *Figure 1—figure supplement 1E*: 200–600 nM; *Figure 1—figure supplement 1F*: 40 nM; MS/MS HEK293T: 100 nM; MS/MS U2OS: 400 nM; *Figure 4F*: 100 nM; *Figures 5G, 6A*: 40 nM; *Figure 6—figure supplement 1A*: 40 nM; *Figure 6H*: 400 nM; *Figure 6—figure supplement 1G*: 400 nM; *Figure 6—figure supplement 2A–C*: 400 nM. |
| CD00509 | TDP1 | 4 µM |
| Etoposide | TOP2 | *Figure 1C*: 3 µM; *Figure 1E*: 0.6–3 µM; *Figure 4F*: 3 µM. |
| Ku-55933 | ATM | Everywhere 9 µM, except in *Figure 4—figure supplement 1H*: 3, 6, or 9 µM, and in *Figure 5G*: 13.5 µM |
| Ku-58948 | PARP1-3 | 4 µM |
| VE-821 | ATR | 3 µM |

## Western blotting and (immuno)staining

For Western blotting, proteins were transferred to either nitrocellulose or PVDF membranes, probed with the indicated antibodies, and imaged in a Bio-Rad ChemiDoc imaging system. For an overview of all antibodies used in this study, see Key resources table.

For (immuno)staining, cells were grown on coverslips, fixed in 2% formaldehyde, permeabilized with 0.1–0.2% Triton-X100, and incubated for 15 min with 0.5% BSA and 0.1% glycine solution in PBS. ProteoStat staining (ENZO, ENZ-51023-KP050) was performed according to the manufacturer's instructions. Primary antibody incubation (see Key resources table) was performed overnight at 4°C. After secondary antibody incubation, cells were stained with Hoechst (Invitrogen, H1399) or DAPI as indicated, and mounted on microscopy slides in Citifluor (Agar Scientific). Cells were observed using a confocal scanning microscope (Leica), and images were processed using ImageJ software (Fiji). The aggresome signature was defined as cells exhibiting both a curved nucleus and perinuclear presence of ProteoStat dye. For DNA repair kinetics experiments, cells were irradiated with 2 Gy (IBL-637 irradiator, CIS Biointernational), fixed at the indicated timepoints post-irradiation, and stained for 53BP1. For the localization experiments of HSPB5, HSP70, and HSP90, U2OS cells were either irradiated with 2 Gy or treated for 24 hr with CPT (400 nM), and left to recover for 48 hr before immunostaining.

## Genotoxic drug treatments

See *Table 1* (and Key resources table) for an overview of the drugs and concentrations used in this study. For genotoxic drug treatments, cells were treated with drugs in the indicated doses. The culture medium was replaced 24 hr after drug treatment, and after another 48 hr (unless explicitly mentioned otherwise) cells were harvested by scraping in PBS, centrifugation, and snap-freezing in liquid nitrogen.

## Differential detergent protein fractionation

Cells were resuspended in ice-cold lysis buffer containing 25 mM HEPES pH 7.4, 100 mM NaCl, 1 mM MgCL$_2$, 1% v/v Igepal CA-630 (#N3500, US Biological), cOmplete EDTA-free protease inhibitor cocktail (Roche Diagnostics), and 0.1 unit/µl benzonase endonuclease (Merck Millipore) and left for 1 hr on ice with intermittent vortexing. Protein content was measured and equalized, and Igepal CA-630 insoluble proteins were pelleted by high-speed centrifugation (21,000 rcf, 45 min, 4°C). Protein pellets were washed with lysis buffer without Igepal CA-630 and redissolved in lysis buffer supplemented with 1% v/v SDS at room temperature (RT) in a Thermomixer R (Eppendorf) at 1200 rpm for 1–2 hr. SDS-insoluble proteins were then pelleted by high-speed centrifugation (21,000 rcf, 45 min). SDS-insoluble protein pellets were washed with lysis buffer without any detergent. For subsequent silver staining, pellets were solubilized in urea buffer (8 M urea, 2% v/v SDS, 50 mM DTT, 50 mM Tris/HCl pH 7.4) overnight at RT in a Thermomixer R (Eppendorf) at 1200 rpm. For subsequent Western blotting, pellets were solubilized in concentrated sample buffer (4% SDS, 125 mM Tris pH 6.8, 100 mM DTT, 10% glycerol, bromophenol blue), boiled for 10 min, and left

overnight RT in a Thermomixer R (Eppendorf) at 1200 rpm. Fractions were separated using SDS-PAGE, imaged using a Bio-Rad ChemiDoc imaging system, and analyzed using ImageJ software (Fiji).

## LC-MS/MS analysis

Samples were reduced (dithiothreitol 25 mM, 37°C, 30 min), alkylated (iodoacetamide 100 mM, RT, 30 min in darkness) and trypsin digested on S-trap columns (Protifi) using the S-Trap micro protocol (https://files.protifi.com/protocols/s-trap-micro-long-4-7.pdf). After elution, samples where dried up on speed-vac and resuspended in 25 µl of 0.1% (v/v) formic acid in water (MS quality, Thermo). Mass spectral analysis was conducted on a Thermo Scientific Orbitrap Exploris. The mobile phase consisted of 0.1% (v/v) formic acid in water (A) and 0.1% (v/v) formic acid in acetonitrile (B). Samples were loaded using a Dionex Ultimate 3000 HPLC system onto a 75 µm × 50 cm Acclaim PepMap RSLC nanoViper column filled with 2 µm C18 particles (Thermo Scientific) using a 120 min LC-MS method at a flow rate of 0.3 µl/min as follows: 3% B over 3 min; 3–45% B over 87 min; 45–80% B over 1 min; then wash at 80% B over 14 min, 80 to 3% B over 1 min and then the column was equilibrated with 3% B for 14 min. For precursor peptides and fragmentation detection on the mass spectrometer, MS1 survey scans ($m/z$ 200–2000) were performed at a resolution of 120,000 with a 300% normalized AGC target. Peptide precursors from charge states 2–6 were sampled for MS2 using Data Dependent Acquisition (DDA). For MS2 scan properties, Higher Energy Collision Dissociation (HCD) was used and the fragments were analyzed in the orbitrap with a collisional energy of 30 %, resolution of 15000, Standard AGC target, and a maximum injection time of 50 ms.

MaxQuant version 1.6.7.0 was used for peptides and protein identification (*Tyanova et al., 2016*) and quantification with a proteomic database of reviewed proteins sequences downloaded from Uniprot (08/17/2020, proteome:up000005640; reviewed:yes). Abbreviated MaxQuant settings: LFQ with minimum peptide counts (razor + unique) ≥ 2 and at least one unique peptide; variable modifications were oxidation (M), acetyl (protein N-term), and phospho (STY); carbamidomethyl (C) was set as a fixed modification with trypsin/P as the enzyme.

ProteinGroup.txt from MaxQuant output was used for protein significance analysis via postprocessing in R (R Core Team 2021): potential contaminant and reversed protein sequences were filtered out, partial or complete missing values in either case or control replicates were imputed (*Dou et al., 2020*) in parallel 100 times, and subsequently averaged $\log_2$-transformed LFQ intensities were used for $t$-tests, including Benjamini–Hochberg-corrected, $p$-adjusted values. $\log_2$ fold change for each protein record was calculated by subtracting the average $\log_2$ LFQ intensity across all replicates in control samples from the average $\log_2$ LFQ intensity across all replicates in case samples. To mitigate imputation-induced artifacts among significant proteins, only significant proteins detected and quantified in at least two replicates were considered: p-adjusted value ≤ 0.05 and, for cases ($\log_2$ fold change ≥ 1, replicates with nonimputed data ≥ 2), or for controls ($\log_2$ fold change ≤ –1, replicates with nonimputed data ≥ 2).

## RNAseq library construction and sequencing

RNA was isolated from cells with the AllPrep DNA/RNA Mini Kit from QIAGEN. RNA concentrations were measured on a NanoDrop. 150 ng of RNA was used for library preparation with the Lexogen QuantSeq 3′ mRNA-Seq Library Prep Kit (FWD) from Illumina. Quality control of the sequencing libraries was performed with both Qubit (DNA HS Assay kit) and Agilent 2200 TapeStation systems (D5000 ScreenTape). All libraries were pooled equimolar and sequenced on a NextSeq 500 at the sequencing facility in the University Medical Center Groningen, Groningen, the Netherlands.

Data preprocessing was performed with the Lexogen QuantSeq 2.3.1 FWD UMI pipeline on the BlueBee Genomics Platform (1.10.18). Count files were loaded into R and analyzed with edgeR *Robinson et al., 2010*. Only genes with >1 counts in at least two samples were included in the analysis. Count data was normalized using logCPM for principal component analysis (PCA). Differential gene expression analysis was performed using the likelihood ratio test implemented in edgeR. Cutoffs of an absolute log fold change > 1 and an FDR-adjusted p-value<0.05 were used to identify significantly differentially expressed genes.

## GO term analyses

For MS/MS, GO term analyses were performed through Cytoscape within Python, with a redundancy cutoff of 0.2. For RNA sequencing, GO term analyses were performed through Metascape (webserver: https://metascape.org) using default settings.

## Radioactive pulse labeling

Radioactive pulse labeling experiments were executed with cells subjected to the same CPT treatment regime as depicted in *Figure 1—figure supplement 1D* (see also *Figure 4D*). After 48 hr of recovery, cells were starved of methionine and cysteine for 30 min DMEM without methionine and cysteine (Gibco), see Key resources table, supplemented with 10% dialyzed FBS (Sigma-Aldrich), 100 units/ml penicillin, and 100 µg/ml streptomycin (Invitrogen). Then, $^{35}$S-met/cys (Hartmann Analytics) pulse labeling was performed for 10–40 min, and immediately after cells were harvested by scraping in PBS, centrifugation, and snap-freezing in liquid nitrogen. Protein fractionation was then performed as described. Autoradiography was performed by running samples on SDS-PAGE gels, gel drying, and placing gels on blank phosphor screens, shielded from light. After 1 week, phosphor screens were imaged using a Cyclone Plus Phosphor Image (Perkin Elmer).

## Quantification of polyglutamine aggregation

24 hr after seeding, stable tetracycline-inducible HTT Q71-GFP-expressing HEK293 cells were treated with the indicated genotoxic drugs listed in *Table 1*, as described. Cell lysis, polyQ filter trap, and immunodetection were performed as described previously (*Kakkar et al., 2016*), and results were analyzed using ImageJ software (Fiji).

## CAG repeat length analysis

DNA was isolated from HTT Q71-GFP-expressing HEK293 cells through MasterPure Complete DNA and RNA Purification Kit (Epicentre) according to the manufacturer's instructions. The CAG repeat length analysis was performed by PCR with 100 ng of DNA in a 10 µl reaction volume containing AmpliTaq Gold Fast PCR Master Mix (Applied Biosystems), and 0.2 µM of both forward (HEK293TQ71F [FAM]: 5'-GAGTCCCTCAAGTCCTTCC-3') and reverse (HEK293TQ71R: 5'-AAACGGGCCCTCTAGACTC-3') primers, flanking the CAG repeat tract. The samples were subjected to an initial denaturation step (95° C, 10 min), 35 amplification cycles (96°C, 15 s; 59.2°C, 15 s; 68°C, 30 s) and a final extension of 72°C, 5 min. PCR was followed by capillary electrophoresis in a ABI3730XL Genetic Analyzer, and results were analyzed through GeneMapper Software V5.0 (both Applied Biosystems).

## Retroviral overexpression of HSPB5

Retrovirus was produced in the Phoenix-AMPHO retroviral packaging cell line using a pQCXIN–HSPB5 vector as described before (*Schepers et al., 2005*). Briefly, HEK293, U2OS wild-type, and *ATM* KO cells were infected in the presence of 5 µg/ml polybrene (Santa Cruz). Cells in which the HSPB5 vector integrated successfully were selected using G418, and HSPB5 overexpression was confirmed via Western blotting.

## Online tools and databases used

For an overview of online databases used in this study, see *Table 2*.

**Table 2.** Online databases used.

| Analysis | Tool/database | Source/weblink |
|---|---|---|
| Supersaturation | Supersaturation database | *Ciryam et al., 2013* |
| Heat-sensitive proteins | Heat-sensitive protein database | *Mymrikov et al., 2017* |
| Stress-granule constituents | RNA granule database | https://rnagranuledb.lunenfeld.ca |
| (Co)chaperone interactions | HSPA1A/HSPA8 client database | *Ryu et al., 2020* |
| | BioGRID PPI database | https://thebiogrid.org |

## Data availability

The MS/MS proteomics data have been deposited to the ProteomeXchange Consortium via the PRIDE partner repository (*Perez-Riverol, 2018*) with the dataset identifier PXD030166. The RNAseq data generated in this study are available through Gene Expression Omnibus with accession number GSE173940. The R code for the MS/MS analysis can be found here (copy archived at swh:1:rev:b-da88adfdacefd6841d80c0c92e92b33b42c9b9c; *LaCavaLab, 2022a*) and here (copy archived at swh:1:rev:1d1711c210a0ac34f09499aa37c46989439ffcbe; *LaCavaLab, 2022b*). For the RNAseq differential expression analysis the R code can be found on github (copy archived at swh:1:rev:e9e5879e270d8788d6f385159e2efcfd49e9c5e0; *Huiting, 2022*).

## Acknowledgements

This work was supported by a Nederlandse organisatie voor Wetenschappelijk Onderzoek (NWO) grant to SB (ALW 824.15.004) and by generous funding from Charity4Brains. This work was partly funded by the National Institute of General Medical Sciences of the National Institutes of Health (NIH) (R01GM126170 to JL); the content is solely the responsibility of the authors and does not necessarily represent the official views of the NIH.

## Additional information

### Competing interests

John LaCava: The other authors declare that no competing interests exist.

### Funding

| Funder | Grant reference number | Author |
| --- | --- | --- |
| Nederlandse Organisatie voor Wetenschappelijk Onderzoek | ALW 824.15.004 | Steven Bergink |
| National Institutes of Health | R01GM126170 | John LaCava |
| Charity4brains | | Steven Bergink |

The funders had no role in study design, data collection and interpretation, or the decision to submit the work for publication.

### Author contributions

Wouter Huiting, Conceptualization, Formal analysis, Investigation, Methodology, Validation, Visualization, Writing – original draft; Suzanne L Dekker, Formal analysis, Investigation, Validation; Joris CJ van der Lienden, Formal analysis, Investigation, Methodology, Validation, Writing – review and editing; Rafaella Mergener, Investigation, Methodology; Maiara K Musskopf, Formal analysis, Software, Visualization; Gabriel V Furtado, Formal analysis, Methodology, Validation, Visualization; Emma Gerrits, Formal analysis, Methodology, Writing – review and editing; David Coit, Formal analysis, Software, Writing – review and editing; Mehrnoosh Oghbaie, Formal analysis, Software; Luciano H Di Stefano, Methodology, Validation, Writing – review and editing; Hein Schepers, Methodology, Writing – review and editing; Maria AWH van Waarde-Verhagen, Suzanne Couzijn, Investigation; Lara Barazzuol, Investigation, Resources, Writing – review and editing; John LaCava, Formal analysis, Methodology, Resources, Software, Supervision, Writing – review and editing; Harm H Kampinga, Conceptualization, Writing – review and editing; Steven Bergink, Conceptualization, Formal analysis, Funding acquisition, Project administration, Resources, Supervision, Writing – original draft, Writing – review and editing

### Author ORCIDs

Wouter Huiting (ID) http://orcid.org/0000-0002-0371-512X
Joris CJ van der Lienden (ID) http://orcid.org/0000-0002-7012-2471
Rafaella Mergener (ID) http://orcid.org/0000-0002-7814-2936

Hein Schepers http://orcid.org/0000-0002-7731-3318
Lara Barazzuol http://orcid.org/0000-0002-6538-2383
John LaCava http://orcid.org/0000-0002-6307-7713
Harm H Kampinga http://orcid.org/0000-0002-8966-8466
Steven Berginkhttp://orcid.org/0000-0002-1142-869X

### Decision letter and Author response
Decision letter https://doi.org/10.7554/eLife.70726.sa1
Author response https://doi.org/10.7554/eLife.70726.sa2

## Additional files

### Supplementary files
• Supplementary file 1. MS/MS datasets of aggregated and whole-cell lysate (WCL) protein fractions.

• Supplementary file 2. RNA-sequencing differential expression analyses.

• Transparent reporting form

### Data availability
LFQ proteomics and RNA-sequencing analyses are uploaded as supplemental tables in an excel format. Raw data has also been deposited in PRIDE and GEO, respectively. Proteomics data are available via ProteomeXchange with identifier PXD030166. The RNAseq data generated in this study are available through Gene Expression Omnibus with accession number GSE173940. The R code for the MS/MS analysis can be found here (copy archived at swh:1:rev:bda88adfdacefd6841d80c0c92e92b-33b42c9b9c) and here (copy archived at swh:1:rev:1d1711c210a0ac34f09499aa37c46989439ffcbe). For the RNAseq differential expression analysis the R code can be found in github (copy archived at swh:1:rev:e9e5879e270d8788d6f385159e2efcfd49e9c5e0).

The following datasets were generated:

| Author(s) | Year | Dataset title | Dataset URL | Database and Identifier |
|---|---|---|---|---|
| Stefano LD, Huiting W, LaCava JP, Bergink S | 2022 | Aggregation under genotoxic conditions in HEK293T and U2OS cells | https://www.ebi.ac.uk/pride/archive/projects/PXD030166 | PRIDE, PXD030166 |
| Huiting W, Dekker SL, van der Lienden JC, Mergener R, Furtado GV, Gerrits E, Musskopf MK, Oghbaie M, Di Stefano LH, van Waarde-Verhagen MA, Barazzuol L, LaCava J, Kampinga HH, Bergink S | 2022 | Genotoxic conditions in HEK293T and U2OS cells | https://www.ncbi.nlm.nih.gov/geo/query/acc.cgi?acc=GSE173940 | NCBI Gene Expression Omnibus, GSE173940 |

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
