## [Editor Report]

This study investigates how different sources of genotoxic stress exacerbate proteome instability, leading to the formation of protein aggregates. These effects are in part caused by an impaired chaperone network and can be relieved by the action of small chaperones with anti-amyloidogenic function. This work functionally connects two fields of research: responses to genotoxic stress and protein aggregation, both of which have fundamental implications for cellular stability and homeostasis during aging. Connecting these two fields sheds new light on basic mechanisms of cell stability and has the potential to help design interventions that buffer both DNA damage responses and proteome stability.

---

## [Decision Letter]

**Decision letter after peer review:**

Thank you for submitting your article "Targeting DNA topoisomerases or checkpoint kinases results in an overload of chaperone systems, triggering aggregation of a metastable subproteome" for consideration by *eLife*. Your article has been reviewed by 3 peer reviewers, including Dario Riccardo Valenzano as Reviewing Editor and Reviewer #1, and the evaluation has been overseen by David Ron as the Senior Editor.

Essential revisions:

All reviewers acknowledge that this study bridges two distinct fields, namely genotoxicity and proteostasis and has several potential important implications. However, there are a few limitations that have been identified, which need to be addressed for this work to be further considered for publication in *eLife*.

One of the main limitations (acknowledged by two reviewers) is that "two different treatments (CPT drug versus ATM genetic knockdown) are compared in two different cellular backgrounds (non-cancerous HEK-293T versus cancerous U2OS cells). This setup makes it difficult to distinguish effects due to the different cellular background or treatments". Additionally, it is not entirely clear what is the model according to which chaperones become overloaded. Do the chaperones get titrated to damaged DNA or in ubiquitin signaling that is on hyperdrive during DNA damage?

The main points that need to be addressed experimentally/analytically for the paper to be considered for publication are the following:

1. Does HSPB5 overexpression have an effect on CPT-induced protein aggregation? Does HSPB5 over-expression also prevent aggregation in HEK293T CPT treated cells? This would help to determine the breath of action of HSPB5 in combating genotoxic stress related protein aggregation.

2. Compare CPT treatment in U2OS versus HEK-293T cells and ATM inhibition by drug treatment versus CPT drug treatment in one cell type. The analysis should clarify results related to SG proteins, heat-sensitive and enrichment for constituents of disease-associated protein aggregates for the ATM KO aggregation set.

3. Are proteins that aggregate upon DNA damage newly synthesized?

Additionally, I invite the authors to address all the points raised by all reviewers in their rebuttal. We hope to see a revised version of this exciting work.

*Reviewer #1:*

This work has the strength of bridging two key fields related with cellular homeostasis and biology of aging: DNA damage response and proteome stability. The authors convincingly show how different sources of DNA damage, such as impaired ATM and ATR responses, as well pharmacologic blockage of topoisomerase 1 and 2, is conducive to increased protein aggregation. These experiments are conducted in two different cell types, proving conservation of the observed effects across systems. The authors show how aggregation-prone proteins aggregate more readily upon DNA damage induction, which lowers the threshold for aggregation. These results are supported by several independent methods, including testing aggregation kinetics in a known aggregation-prone protein (Hungtingtin exon 1) after DNA damage induction.

Mechanistically, the authors manage to connect lowered aggregation threshold after DNA damage to chaperones dysfunctional networking, which can be buffered by induction of small chaperones, including HSPB5, a known aggregation suppressor.

Throughout the manuscript the claims are well balanced and the authors do a great job in acknowledging potential limitations of their findings, as well as giving credit to previously published work with results that in part overlap and support some parts of their findings.

Congratulations on a very well conducted and written study. I had a great time reading it.

*Reviewer #2:*

Defects in the DNA-damage response (DDR) are the cause of several human genetic syndromes, commonly characterised by neurodegeneration and cancer predisposition, such as Ataxia telangiectasia (A-T), caused by inactivation of ATM, a key regulator of the DDR. Previous studies have linked ATM deficiency to increased protein aggregation. The current manuscript expands the connection between DNA damage and protein aggregation by analysing additional genotoxic conditions and identifying the proteome susceptible to aggregation upon DNA damage. Moreover, the authors show that modulation of chaperone levels affects the extent of DNA-damage-induced aggregation, suggesting that increased aggregation upon DNA damage is likely a result of an overloaded chaperone system.

The specific observations are:

– The authors found that genotoxic conditions, such as treatment with topoisomerase I poison camptothecin (CPT), cause an increase in protein aggregation.

– The authors found that aggregating proteins produced by ATM loss and CPT treatment are different, probably because of cell-type proteome differences.

– In addition, the authors found that aggregating proteins upon CPT treatment are in general supersaturated and prone to aggregate.

– The authors showed that genotoxic conditions accelerate the aggregation of expanded polyQ protein aggregation models without increasing the number of CAG repeats.

– The authors found that CPT treatment reduces the solubility of multiple chaperones and causes changes in chaperone gene expression.

– Also, authors found that HSP70 inhibitor increased CPT-induced protein aggregation while the overexpression of HSPB5 reduced protein aggregation observed in ATM deficient cells.

The results broadly support the descriptive conclusions made in the manuscript.

Importantly, the study links two rather distinct fields, namely genotoxicity and proteostasis, thus bring out novel thoughts and datasets.

Some points for further consideration are:

(1) Previous study showed that proteasome-mediated protein degradation is impaired in ATM deficient cells (PMID: 21298066) and defects on ATM have shown increased protein aggregation (PMID: 29317520 and 16189515). In addition, deficiencies in ATM and the related protein MRE11 have also been shown to cause protein aggregation driven by poly-ADP-ribose polymerases (PARPs) hyperactivation (PMID: 33571423). The proposed manuscript expands the association between DNA damage and protein insolubility by describing additional genotoxic insults also cause protein aggregation. The manuscript performs various different analyses with lists of aggregated proteins which may prove useful in linking this study to neurodegeneration and ageing. On the mechanistic side, however, the underlying causes that link DNA damage and protein aggregation remain unknown as detailed below.

(i) The various descriptive analyses to characterise protein aggregation upon genotoxic conditions are convincing and complementary. However, the 24h genotoxic treatment followed by 48h additional hours before studying aggregation introduces complex effects of these stressful conditions. The long time gap between DNA damage and sampling for aggregation makes interpretation of direct causality of DNA damage on aggregation hard. E.g. could there be a cell-cycle checkpoint activated/ mitochondria/ ER defects introduced as a consequence of DNA damage that ultimately lead to protein aggregation?

(ii) Previous studies have shown that ATM can be independently activated by oxidative stress caused by reactive oxidative species (ROS) (PMID: 20966255) and Double-strand Breaks by a Mre11-dependent mechanism. Defects on ROS-dependent activation of ATM cause dysregulation of protein homeostasis (PMID: 29317520), suggesting ROS increase and PARP hyperactivation as the source of neurotoxicity observed in Ataxia Telangiectasia (A-T) disease. However, mutations in Mre11 also generate cerebellar neurodegeneration and protein aggregation by a ROS-independent mechanism (PMID: 33571423). If would be especially interesting to assess if protein aggregation produced by additional genotoxic treatments used in this study (ATR inhibition, topoisomerase poisoning) depends on ROS accumulation and PARP hyperactivation as well. In addition, it would be interesting to evaluate the activation of the major regulator of protein homeostasis (HSF1) upon the different insults. Furthermore, it would be clarifying to analyse if genotoxic-induced protein aggregation depends on nascent translation to offer more mechanistic details. E.g. it has been shown that acute proteotoxic stress causes nascent polypeptides to get ubiquitinated. It would be interesting to know whether the proteins that aggregate upon DNA damage are newly synthesized.

(2) Along the manuscript two different models are used (ATM loss in U2OS and CPT treatment in HEK29T cells). Comparisons between the protein aggregation induced by the two genotoxic conditions are complicated since they are distinct cell lines and should be avoided. Furthermore, author should explain why only some models are used in some experiments, since for example, it might be interesting to assess if HSPB5 overexpression has an equivalent effect on CPT-induced protein aggregation.

(3) The accelerated aggregation of polyQ-GFP upon genotoxic conditions results are clear, however, authors should include whole-cell extract (WCE) controls in these experiments, since the level of aggregation is completely dependent on the expression levels of the protein and the used treatments can alter expression levels. For instance, this can be observed in Figure 4G (comparing ATM and TOP1 inhibition).

(4) The authors argue that genotoxic conditions cause the aggregation of supersaturated proteins. Although, this might be true in the case of CPT-treated HEK293T, it doesn't explain the protein aggregation observed in ATM knockout U2OS cells. Author should discuss this in further details and offer more convincing observation to strengthen their conclusions.

(5) The authors state that upon CPT treatment the expression of certain chaperone was altered, this result doesn't seem to be well supported by the volcano plots shown in Figure 5E and 5F. The authors should discuss the transcriptional changes results from a broad perspective since, clearly, chaperone-encoding genes are not the most differentially expressed genes.

(6) The chaperone overload model to explain DNA-damage-induced aggregation is an interesting interpretation. However it is not clear if HSP70 inhibition/ HSPB5 overexpression alter the manner in which aggregated proteins are handled by cells. Thus it is hard to argue whether chaperone overload is the reason for the aggregation in the first place or the experimental manipulation of chaperones (HSP70 inhibition) lowers the ability of cells to manage aggregated proteins. If chaperone overload is the reason of aggregation, what is causing chaperones to be overloaded? Do the chaperones get titrated to damaged DNA or in ubiquitin signaling that is on hyperdrive during DNA damage?

Long treatment time used allow cells to pass through several DNA replication cycles, so the authors could evaluate whether the observed protein aggregation upon genotoxic treatments depends on DNA replication.

Previous studies in yeast have proposed that deficiency in ribosomal RNA and proteins synthesis lead to proteotoxic stress (PMID: 30843788) and defects on DNA topoisomerase I have been shown to cause similar consequences (PMID: 1124783). Therefore, it would be worth assessing whether rRNA stress/instability could explain the increased protein aggregation observed upon genotoxic insults again to offer mechanistic insights.

The Figure 4G is not cited in the manuscript.

WCE levels of HSPB5 overexpression should also be shown in corresponding experiments (Figure 6C, G), as well as the total expression of FUS in WT and ATM deficient cells by the same reasons as mentioned above.

Also, the authors should provide the list of the RNAseq analysis results showing the changes in gene expression and the statistical information.

*Reviewer #3:*

Protein instability with subsequent widespread protein aggregation occurs during aging, in neurodegenerative diseases and in response to various stresses. Previous work from different groups identified widespread protein aggregation linked to impairment of the DNA damage response. This work sets out to investigate the impact on protein aggregation following treatment with a variety of drugs targeting the DNA damage response.

Strengths:

The study provides an in depth comparison of protein aggregation in two different cell lines in response to impairment of DNA damage repair. Extensive bioinformatics analysis reveals the characteristics of the aggregated fraction after treatment of CPT targeting topoisomerase 1. In particular, they show an enrichment of LLPS prone proteins and proteins previously identified with neurodegenerative disease aggregates. Importantly with a comprehensive analysis of whole proteome and insoluble proteome mass spectrometry data combined with RNAseq data, they expose changes in the chaperone system. Using an over-expression screen, they find that increasing levels of small heat shock protein HSPB5 reduces widespread protein aggregation following genotoxic stress related to ATM knockout.

Weaknesses:

While the study benefits from a comprehensive omics and bioinformatics analysis, the main experimental setup is questionable: two different treatments (CPT drug versus ATM genetic knockdown) are compared in two different cellular backgrounds (non-cancerous HEK-293T versus cancerous U2OS cells). This setup makes it difficult to distinguish effects due to the different cellular background or treatments.

Line 125: The claim that "different drug treatments drive the aggregation of a similar set of proteins (Figure 1C)" is not sufficiently justified. This claim is based on the analysis of only three proteins and a densitometry analysis of the stained insoluble proteome on an SDS-PAGE gel. Also the claim is not substantiated by an overlap between the proteins in the increased aggregation fraction in CPT versus ATM KO treatments (Figure 2E). The previous study by Lee et al., 2021 referred to in the introduction, provides a more relevant comparison showing the strong overlap of proteins in the insoluble fraction in the same cellular background with loss of MRE11 or ATM.

Line 135: “we identified a total of 1826 aggregated proteins across U2OS wild-type and ATM KO cells” and line 115 “We picked up a total of 983 aggregated proteins”. These numbers include proteins identified in only one replicate of the aggregated fractions from either case or control. I would advise to state the number of proteins identified in >1 repeats from either case or control.

Line 161-163: “we examined which proteins already aggregate in the background of untreated HEK293T and U2OS cells (Benjamini-Hochberg corrected p>0.05, identified in >1 repeats of both case and control) (Figure 2A, D). The definition of the “Background aggregation” is unclear and potentially misleading. If only non-significant proteins in over one repeat in both case and control are included, then the background aggregation set does not include proteins identified only in untreated. Also proteins with increased aggregation that also aggregate in the control conditions are not included in the background set. This explains why there is no overlap between increased aggregation and background aggregation in Figure 2E.

Line 255: “we conclude that both CPT-treatment and a loss of ATM further exacerbate the aggregation of LLPS-prone and supersaturated proteins, in a cell-type dependent manner”. As one cell is cancerous and the other not, it is unclear if any conclusions can be made related to cell-type.

HSPB5 is significantly up-regulated in U2OS ATM KO but not in HEK293T CPT treated cells. Does HSPB5 over-expression also prevent aggregation in HEK293T CPT treated cells? This would help to determine the breath of action of HSPB5 in combating genotoxic stress related protein aggregation.

Additional experiments to strengthen the science:

I recommend comparing CPT treatment in U2OS versus HEK-293T cells and ATM inhibition by drug treatment versus CPT drug treatment in one cell type. It would be interesting to see the results related to SG proteins, heat-sensitive and enrichment for constituents of disease-associated protein aggregates for the ATM KO aggregation set. Also it would be important to show that HSPB5 overexpression alleviates protein aggregation in HEK293T CPT treated cells.

---

## [Author Response]

Essential revisions:All reviewers acknowledge that this study bridges two distinct fields, namely genotoxicity and proteostasis and has several potential important implications. However, there are a few limitations that have been identified, which need to be addressed for this work to be further considered for publication in eLife.One of the main limitations (acknowledged by two reviewers) is that “two different treatments (CPT drug versus ATM genetic knockdown) are compared in two different cellular backgrounds (non-cancerous HEK-293T versus cancerous U2OS cells). This setup makes it difficult to distinguish effects due to the different cellular background or treatments”.

We agree with the reviewers, and now added two additional MS/MS and RNAseq datasets, namely HEK293T cells treated with ATM inhibitor and U2OS cells treated with CPT. These datasets allow us now to compare different treatments within a cell line, and a single stress across the two cell lines. These comparisons support our earlier conclusions, and in our opinion these datasets definitely improved the manuscript, as we think that the study certainly became more mature, and the conclusions more solid.

In addition, we made use of the opportunity to also improve the statistical analysis of our MS/MS data. Previously, we relied on a 1-seed imputation method to impute missing data. However, we realized that a single random sample from a given distribution can render outliers, especially when such a sampling process is repeated for a number of missing values. We therefore opted to switch to a more accurate 100-seed imputation method, in which for each missing value 100 samples are generated using different seeds and subsequently averaged. This ensures that randomly distributed outliers have a minimal impact on the imputed values. All details regarding statistical analyses are available in the Materials and methods section.

Note that because of these improvements, many figures have changed.

Additionally, it is not entirely clear what is the model according to which chaperones become overloaded. Do the chaperones get titrated to damaged DNA or in ubiquitin signaling that is on hyperdrive during DNA damage?

Our data does not point to a titration of chaperones to DNA damage sites. For example, HSPB5 overexpression does not impact the kinetics of γ irradiation-induced DNA damage sites. To strengthen this argument, we now provide additional data showing that none of the chaperones that we tested is present at DNA damage sites caused either by CPT or by γ irradiation (Figure 6 —figure supplement 2).

In our manuscript we hypothesize that in certain genotoxic backgrounds chaperones become overloaded because of an increased cellular demand for chaperone capacity emanating from the proteome, affecting mostly supersaturated, LLPS-prone client proteins that subsequently aggregate. Importantly, this includes the aggregation of putative clients of those chaperones that also co-aggregate. This suggests that the aggregation of certain chaperones may in cases of high levels of genotoxic stress further accelerate the aggregation of their clients/substrates in a vicious cycle.

This study bridges genotoxic stress with protein aggregation. Uncovering further molecular details regarding the initial trigger of the increased cellular demand for chaperone capacity goes beyond the aim of this study – they will need to be elucidated by future studies on this newly described phenomenon. In the manuscript we did rephrase parts of the Discussion (lines 519-529 and 541-543) but at this point, we feel that a further in-depth discussion of this would quickly become too speculative, so we refrained from it in the manuscript. Having said that, in our opinion the link between DNA damage and liquid liquid phase separation is an intriguing option. PARylation is known to decorate various types of (single strand) DNA damage. PARylation also plays a key role in the regulation of LLPS processes, and the addition of PAR to a substrate has been hypothesized to initiate and accelerate LLPS, and increase the risk of progressing into a solid-like state. The fact that proteins that aggregate after genotoxic stress are frequently LLPS-prone therefore hints at the existence of a coupling between DNA damage and PARP hyperactivation occurring throughout the cell. This could lead to an increased demand for chaperone capacity regulating LLPS processes. It is possible that besides PARylation, other protein modifications would play a role here as well.

It might indeed be possible that ubiquitin signaling plays a role as it is a crucial part of the DNA Damage Response. However, we want to emphasize that our analysis reveals that there is no direct correlation between an increase in protein level and the increased aggregation (Figure 4A and Figure 4 —figure supplement 1A). In other words, the proteins that aggregate do not do so because their levels were altered, because of, for example, an impaired UPS-mediated degradation. However, we certainly do not want to rule out a role for ubiquitin-mediated signaling.

The main points that need to be addressed experimentally/analytically for the paper to be considered for publication are the following:1. Does HSPB5 overexpression have an effect on CPT-induced protein aggregation? Does HSPB5 over-expression also prevent aggregation in HEK293T CPT treated cells? This would help to determine the breath of action of HSPB5 in combating genotoxic stress related protein aggregation.

We tested both questions. Overexpression of HSPB5 can suppress the aggregation induced by CPT in U2OS cells (Figure 6H, I). As requested, we now also generated HEK293 cells stably overexpressing HSPB5, but found that in those cells CPT still triggers enhanced aggregation (Figure 6 —figure supplement 1G). This is perhaps not surprising as HSPB5 is barely present in HEK293 cells, and it is also not upregulated under the conditions of genotoxic stress used in our study. It has been suggested that small heat shock chaperones (including HSPB5) operate together in hetero-dimers and hetero-oligomers, and that in vivo they depend on other chaperone systems to efficiently counteract aggregation. Our findings indicate that the rewiring of chaperone systems (and by extension, the possibility of rescuing aggregation through overexpression of in this case HSPB5) is tailored to each cell. We changed our conclusions accordingly (in both the Discussion (lines 490-500) and the Abstract (lines 13-14)).

2. Compare CPT treatment in U2OS versus HEK-293T cells and ATM inhibition by drug treatment versus CPT drug treatment in one cell type. The analysis should clarify results related to SG proteins, heat-sensitive and enrichment for constituents of disease-associated protein aggregates for the ATM KO aggregation set.

As mentioned above, we now added two extra datasets, which allowed us to compare the two treatments in the two different cell lines. We also included these in the subsequent analyses, including the enrichment for proteins that aggregate after HS, constituents of stress granules, and in certain aggregation-diseases (Figure 4B, C and E, Figure 4 —figure supplement 1C, D). Briefly, we find that aggregation induced by a loss of ATM function also overlaps with the aggregation in HS, SG components, and certain disease aggregates, and hence further support our earlier conclusions.

3. Are proteins that aggregate upon DNA damage newly synthesized?

We performed pulse experiments with radioactive (^35^S) methionine and cysteine in combination with our aggregation assay. This revealed a strong reduction in protein synthesis after CPT treatment (as has been shown before for other forms of DNA damage). Despite this reduction in protein synthesis, we can clearly measure that newly synthesized proteins indeed still aggregate potently (Figure 4E). That being said, the relative contribution of newly synthesized proteins to the aggregated fraction is not significantly altered after genotoxic stress (Figure 4D). This indicates that the enhanced aggregation cannot be (solely) explained by an increased aggregation of specifically newly synthesized proteins.

Reviewer #2:[….]Some points for further consideration are:(1) Previous study showed that proteasome-mediated protein degradation is impaired in ATM deficient cells (PMID: 21298066) and defects on ATM have shown increased protein aggregation (PMID: 29317520 and 16189515). In addition, deficiencies in ATM and the related protein MRE11 have also been shown to cause protein aggregation driven by poly-ADP-ribose polymerases (PARPs) hyperactivation (PMID: 33571423). The proposed manuscript expands the association between DNA damage and protein insolubility by describing additional genotoxic insults also cause protein aggregation. The manuscript performs various different analyses with lists of aggregated proteins which may prove useful in linking this study to neurodegeneration and ageing. On the mechanistic side, however, the underlying causes that link DNA damage and protein aggregation remain unknown as detailed below.(i) The various descriptive analyses to characterise protein aggregation upon genotoxic conditions are convincing and complementary. However, the 24h genotoxic treatment followed by 48h additional hours before studying aggregation introduces complex effects of these stressful conditions. The long time gap between DNA damage and sampling for aggregation makes interpretation of direct causality of DNA damage on aggregation hard. E.g. could there be a cell-cycle checkpoint activated/ mitochondria/ ER defects introduced as a consequence of DNA damage that ultimately lead to protein aggregation?

DNA damage, including treatment with topoisomerase poisons, is known to elicit a cell cycle checkpoint arrest. One of the primary defects in ATM or ATR-compromised cells is that they are impaired in eliciting this checkpoint response. The observation that aggregation is induced after when lacking functional checkpoint kinases suggests that the checkpoint activation per se is not necessary to induce aggregation under these circumstances. Still, it could be possible that checkpoint activation influences the aggregation process. To correctly and fully assess the role of cell cycle checkpoints would be a separate study in itself that goes beyond the scope of the present study.

Mitochondrial and ER-resident proteins are among the proteins that aggregate, as for example becomes clear from the GO-term analyses of aggregating proteins in HEK293T cells treated with CPT (Figure 2 —figure supplement 1D-I). However, this is not consistent between the two different treatments, and between the two cell lines, suggesting that mitochondrial or ER damage does not play a primary role in aggregation induced by genotoxic stress.

(ii) Previous studies have shown that ATM can be independently activated by oxidative stress caused by reactive oxidative species (ROS) (PMID: 20966255) and Double-strand Breaks by a Mre11-dependent mechanism. Defects on ROS-dependent activation of ATM cause dysregulation of protein homeostasis (PMID: 29317520), suggesting ROS increase and PARP hyperactivation as the source of neurotoxicity observed in Ataxia Telangiectasia (A-T) disease. However, mutations in Mre11 also generate cerebellar neurodegeneration and protein aggregation by a ROS-independent mechanism (PMID: 33571423). If would be especially interesting to assess if protein aggregation produced by additional genotoxic treatments used in this study (ATR inhibition, topoisomerase poisoning) depends on ROS accumulation and PARP hyperactivation as well.

We tested both the dependency of ROS and PARP in the aggregation process using the inhibitors NAC and Ku-58948, respectively. The results for blocking ROS with NAC yielded inconsistent results (data therefore not included in the manuscript). The effects of blocking PARP were striking: PARP inhibition can reduce CPT-induced aggregation in HEK293T cells (Figure 1 —figure supplement 1F, G), which is highly reminiscent of the results published before for ATM deficient cells (PMID: 33571423). This implies that PARP (hyper)activation may play a role in genotoxic stress-induced aggregation.

In addition, it would be interesting to evaluate the activation of the major regulator of protein homeostasis (HSF1) upon the different insults.

We tested the activation of HSF1 in both ATM-compromised cells and after CPT treatment. A partial activation of HSF1 can be observed after CPT-treatment; however, after ATM inhibition this is less prominent. This is in line with an overall less strong aggregation response in the case of ATM inhibition (Figure 1C, Figure 5 —figure supplement 1A).

Furthermore, it would be clarifying to analyse if genotoxic-induced protein aggregation depends on nascent translation to offer more mechanistic details. E.g. it has been shown that acute proteotoxic stress causes nascent polypeptides to get ubiquitinated. It would be interesting to know whether the proteins that aggregate upon DNA damage are newly synthesized.

We tested whether newly synthesized proteins aggregate in cells subjected to CPT and find that they do, despite a strong reduction in protein synthesis (Figure 4D). However, the relative contribution of newly synthesized proteins to aggregation appears to be unchanged (Figure 4D). This indicates that the enhanced aggregation occurring in CPT-treated cells cannot be (solely) explained by an enhanced aggregation of newly synthesized proteins.

(2) Along the manuscript two different models are used (ATM loss in U2OS and CPT treatment in HEK29T cells). Comparisons between the protein aggregation induced by the two genotoxic conditions are complicated since they are distinct cell lines and should be avoided.

The reviewer is absolutely correct and we now added two extra MS/MS and RNAseq datasets that support our earlier conclusions.

Furthermore, author should explain why only some models are used in some experiments, since for example, it might be interesting to assess if HSPB5 overexpression has an equivalent effect on CPT-induced protein aggregation.

This has largely changed with the addition of the new datasets. With regards to the latter: we now show that overexpression of HSPB5 can indeed suppress CPT-induced aggregation in U2OS cells (Figure 6H, I).

(3) The accelerated aggregation of polyQ-GFP upon genotoxic conditions results are clear, however, authors should include whole-cell extract (WCE) controls in these experiments, since the level of aggregation is completely dependent on the expression levels of the protein and the used treatments can alter expression levels. For instance, this can be observed in Figure 4G (comparing ATM and TOP1 inhibition).

The WCE controls can be found in Figure 4 —figure supplement 1G and H.

CPT and Etop indeed lead to higher levels in the WCE of the GFP-Q71 tetracycline-inducible stable cell line we used. However, it is important to note that the accelerated aggregation cannot be explained by this increase alone. For example, ATM and ATR inhibitors do not have this effect, while still inducing polyQ-GFP aggregation (Figure 4F, Figure 4 —figure supplement 1G and I). Low doses of Etoposide do not increase the total levels of GFP-Q71 (Figure 4 —figure supplement 1H), while these doses do lead to an accelerated aggregation (Figure 4G).

(4) The authors argue that genotoxic conditions cause the aggregation of supersaturated proteins. Although, this might be true in the case of CPT-treated HEK293T, it doesn’t explain the protein aggregation observed in ATM knockout U2OS cells. Author should discuss this in further details and offer more convincing observation to strengthen their conclusions.

Our new data and improved analysis provide additional clarification on this point. We now show that ATM treatment of HEK293T cells also triggers the enhanced aggregation of supersaturated, LLPS-prone proteins. Importantly, many of the proteins that aggregate after CPT treatment or ATM-inhibition in HEK293T cells are already subject to aggregation in U2OS cells under control conditions and are present in what we refer to as the ‘baseline aggregation’ (Figure 2F). *ATM* KO and CPT treatment in U2OS cells still trigger additional aggregation. These additionally aggregating proteins appear indeed less supersaturated than those in HEK293T cells, and in fact the difference with the U2OS NIA (i.e. non-aggregating proteins) is no longer significant for CPT-treated cells (Figure 3 —figure supplement 1M, N). Aggregating proteins in *ATM* KO cells are LLPS-prone, for proteins that aggregate after CPT-treatment this is less prominent. Importantly, as in the HEK293T experiments, proteins that aggregate under these two conditions are both still enriched for heat-sensitive proteins, stress granule constituents (Figure 4 —figure supplement 1C, D), and chaperone clients (Figure 5B, C and Figure 5 —figure supplement 2). This pattern suggests that different cell lines have different “layers” of vulnerable proteins, depending on their ground state of protein homeostasis. In HEK293T cells this ground state is high, putting only highly vulnerable proteins (high supersaturation, high LLPS propensity) under threat of aggregation upon genotoxic stress. In U2OS cells this ground state is much lower, and many of the most vulnerable proteins are already subject to aggregation. Genotoxic stress in U2OS therefore elicits the aggregation of proteins that are less well-predicted by supersaturation and LLPS-proneness. This is visualized in Figure 4 —figure supplement 1B, which we adapted to illustrate this point more clearly.

(5) The authors state that upon CPT treatment the expression of certain chaperone was altered, this result doesn’t seem to be well supported by the volcano plots shown in Figure 5E and 5F.

In the previous manuscript we referred to both transcript and protein levels while these changes were more apparent at the transcript levels, we agree that this was confusing. We changed this in the text (lines 370-375). Importantly, the improved analysis of our data and the inclusion of the two new datasets (Figure 5 and in Figure 5 —figure supplement 1) still allow us to conclude that (functional) loss of ATM or CPT treatment does result in a rewiring of the chaperone network (although more subtle for ATM inhibition in HEK293T cells) in a cell line and stress-specific manner.

The authors should discuss the transcriptional changes results from a broad perspective since, clearly, chaperone-encoding genes are not the most differentially expressed genes.

We performed a GO-term analysis (Figure 3 —figure supplement 2) and added an accompanying paragraph in the text that discusses the transcriptional changes from a broader perspective (lines 250-256). We kept this concise however, as we found no obvious overlap between stress conditions or global transcriptional changes that would clearly provide additional explanations as to the forces that drive aggregation.

(6) The chaperone overload model to explain DNA-damage-induced aggregation is an interesting interpretation. However it is not clear if HSP70 inhibition/ HSPB5 overexpression alter the manner in which aggregated proteins are handled by cells. Thus it is hard to argue whether chaperone overload is the reason for the aggregation in the first place or the experimental manipulation of chaperones (HSP70 inhibition) lowers the ability of cells to manage aggregated proteins.

The “overload of chaperones-model” that we propose is strengthened by our new data, and is based on the following observations: (1) Chaperone clients are significantly enriched in the aggregating fractions (Figure 5B and C). (2) Multiple (co)chaperones aggregate themselves (Figure 5A and Figure 5 —figure supplement 1A-E). (3) Putative clients of those (co)chaperones that aggregate are overrepresented in the list of aggregating proteins (Figure 5D and Figure 5 —figure supplement 2). (4) The inhibition of HSP70, at a dose which is not causing aggregation under normal circumstances, increases the aggregation after CPT even more (Figure 6 —figure supplement 1A). (5) Genotoxic stress leads to a partial activation of HSF1 over time. (6) HSPB5 is upregulated in U2OS cells subjected to genotoxic stress (Figure 5 —figure supplement 1G-H), and overexpression of HSPB5 is able to fully rescue the increased aggregation both after CPT-treatment and after a loss of *ATM* (Figure 6B, C, H, I). Together, these data confirm the prominent role of chaperones in the genotoxic stress-induced aggregation process.

If chaperone overload is the reason of aggregation, what is causing chaperones to be overloaded?

This is an intriguing but difficult question to answer. In our manuscript we focused mainly on the reason why proteins aggregate after genotoxic stress. We reveal that supersaturated proteins that are LLPS-prone aggregate due to an overload of chaperones, which is reminiscent of aggregation after canonical protein stress such as HS. We want to point out that also under these conditions the exact molecular details are not well understood. Our data leads us to hypothesize that the overload of chaperone systems caused by (transient) genotoxic stress is caused by an increased demand for chaperone capacity. This likely includes an increased demand for preventing aberrant LLPS, and may very well also include a more global increase in folding/conformational maintenance demands. We have clarified this in the Discussion (lines 519-529 and 541-543).

Do the chaperones get titrated to damaged DNA or in ubiquitin signaling that is on hyperdrive during DNA damage?

Importantly, HSPB5 overexpression has no impact on the formation and resolution of γ radiation-induced DNA damage foci (Figure 6 —figure supplement 6D, E). We have now also tested a few chaperones that are important for our study (HSPB5, HSP70 and HSP90), and found that none of these accumulate on DNA damage sites, neither on CPT-induced DNA damage nor on γ irradiation-induced DNA damage (Figure 6 —figure supplement 2).

Ubiquitin signaling is an important part of the DNA damage response. Therefore, it might be possible that it plays a role in the accelerated aggregation we observe. How ubiquitin signaling and chaperone function relate is an interesting topic which is not that well understood in general. However, to our knowledge there is no indication that DNA damage-induced ubiquitylation results in a lack of available or “free” ubiquitin that is necessary for other processes.

Long treatment time used allow cells to pass through several DNA replication cycles, so the authors could evaluate whether the observed protein aggregation upon genotoxic treatments depends on DNA replication.

Although some of the higher CPT doses we used elicit a strong checkpoint response which arrested the cells (not shown), thereby minimizing the rounds of replication, we cannot rule out that replication could play an (exacerbating) role in the increased aggregation. To elucidate in how far replication would contribute is in our opinion beyond the scope of the current study.

Previous studies in yeast have proposed that deficiency in ribosomal RNA and proteins synthesis lead to proteotoxic stress (PMID: 30843788) and defects on DNA topoisomerase I have been shown to cause similar consequences (PMID: 1124783). Therefore, it would be worth assessing whether rRNA stress/instability could explain the increased protein aggregation observed upon genotoxic insults again to offer mechanistic insights.

Our RNAseq analysis nor our MS analysis hinted to problems in rRNA synthesis or ribosomal proteins to aggregate more under genotoxic stress conditions. This certainly does not mean that problems in protein synthesis do not play a role, and in fact we now added data showing that newly synthesized proteins are present in the CPT-induced aggregation fraction. However, as explained above, the relative aggregation of newly synthesized proteins is not significantly altered in cells exposed to genotoxic stress.

The Figure 4G is not cited in the manuscript.

We corrected this.

WCE levels of HSPB5 overexpression should also be shown in corresponding experiments (Figure 6C,G), as well as the total expression of FUS in WT and ATM deficient cells by the same reasons as mentioned above.

We have added a blot of the same lysates probed for HSPB5. In the MS/MS experiments we verified that the levels of FUS are not altered in U2OS *ATM* KO cells (see also Supplemental Table 1).

Also, the authors should provide the list of the RNAseq analysis results showing the changes in gene expression and the statistical information.

These data are available in Supplemental Table 2, and through Gene Expression Omnibus at https://www.ncbi.nlm.nih.gov.geo with accession number GSE173940

Reviewer #3:Protein instability with subsequent widespread protein aggregation occurs during aging, in neurodegenerative diseases and in response to various stresses. Previous work from different groups identified widespread protein aggregation linked to impairment of the DNA damage response. This work sets out to investigate the impact on protein aggregation following treatment with a variety of drugs targeting the DNA damage response.Strengths:The study provides an in depth comparison of protein aggregation in two different cell lines in response to impairment of DNA damage repair. Extensive bioinformatics analysis reveals the characteristics of the aggregated fraction after treatment of CPT targeting topoisomerase 1. In particular, they show an enrichment of LLPS prone proteins and proteins previously identified with neurodegenerative disease aggregates. Importantly with a comprehensive analysis of whole proteome and insoluble proteome mass spectrometry data combined with RNAseq data, they expose changes in the chaperone system. Using an over-expression screen, they find that increasing levels of small heat shock protein HSPB5 reduces widespread protein aggregation following genotoxic stress related to ATM knockout.Weaknesses:While the study benefits from a comprehensive omics and bioinformatics analysis, the main experimental setup is questionable: two different treatments (CPT drug versus ATM genetic knockdown) are compared in two different cellular backgrounds (non-cancerous HEK-293T versus cancerous U2OS cells). This setup makes it difficult to distinguish effects due to the different cellular background or treatments.

The reviewer is absolutely correct and we now added two more datasets of MS/MS and RNAseq that complement the previous two. Comparing ATM inhibition and CPT treatment within and between HEK293 and U2OS cells confirmed our previous conclusions that different cell lines have different ground states of protein homeostasis, meaning that the additional proteins that aggregate upon genotoxic stress can differ amongst cell lines.

Line 125: The claim that “different drug treatments drive the aggregation of a similar set of proteins (Figure 1C)” is not sufficiently justified. This claim is based on the analysis of only three proteins and a densitometry analysis of the stained insoluble proteome on an SDS-PAGE gel. Also the claim is not substantiated by an overlap between the proteins in the increased aggregation fraction in CPT versus ATM KO treatments (Figure 2E).

Our new data revealed that this claim was indeed not fully justified, and we therefore adapted the text accordingly.

The previous study by Lee et al., 2021 referred to in the introduction, provides a more relevant comparison showing the strong overlap of proteins in the insoluble fraction in the same cellular background with loss of MRE11 or ATM.Line 135: "we identified a total of 1826 aggregated proteins across U2OS wild-type and ATM KO cells" and line 115 "We picked up a total of 983 aggregated proteins". These numbers include proteins identified in only one replicate of the aggregated fractions from either case or control. I would advise to state the number of proteins identified in >1 repeats from either case or control.

The reviewer is indeed correct in that the total number of aggregating proteins that was picked up is not that meaningful. We therefore removed the comment altogether.

Line 161-163: "we examined which proteins already aggregate in the background of untreated HEK293T and U2OS cells (Benjamini-Hochberg corrected p>0.05, identified in >1 repeats of both case and control) (Figure 2A, D). The definition of the "Background aggregation" is unclear and potentially misleading. If only non-significant proteins in over one repeat in both case and control are included, then the background aggregation set does not include proteins identified only in untreated. Also proteins with increased aggregation that also aggregate in the control conditions are not included in the background set. This explains why there is no overlap between increased aggregation and background aggregation in Figure 2E.

We apologize for not being clear here. We previously indeed defined ‘Background aggregation’ as those proteins that aggregated unchanged in case versus control. They had to be picked up at least twice in case or control. Our aim was to group those proteins that aggregated to a similar extent within a cell line, regardless of the presence or absence of genotoxic stress. We did this because we wanted to investigate if the (physicochemical) properties of those proteins are different from those that aggregate (more) under genotoxic stress. We understand that the term ‘Background aggregation’ may in itself have caused some confusion, and so we changed it to ‘Baseline aggregation’.

We want to emphasize that taking baseline aggregation into account proved critical to our study, as it revealed that genotoxic stress-induced aggregation depends mostly on the cell line-intrinsic threshold of aggregation. We believe that our study is thereby also highly relevant for studies of protein aggregation in general. It illustrates the importance of looking at what we call ‘baseline aggregation’, for example in the context of proteinopathies – this is currently not often done. In other words, the fact that specific proteins aggregate in one cell but not in another may not necessarily mean that different processes are at work, but could very well reflect cell-to-cell differences in the ground state of protein homeostasis.

Of note: our additional MS/MS experiments now allow us to use a more strict cut-off for this group of proteins. Baseline aggregation for HEK293T and U2OS cells is now defined as those proteins that aggregate to a similar extent regardless of stress (Benjamini-Hochberg corrected p>0.05, identified in >1 repeats of both case and control), and that do so in both experiments performed for that cell line. We clarified this in the text (lines 156-164), and for further clarification, we also added a sheet in Supplemental Table 1 that explains how we defined each (non-)aggregating fraction.

Line 255: "we conclude that both CPT-treatment and a loss of ATM further exacerbate the aggregation of LLPS-prone and supersaturated proteins, in a cell-type dependent manner". As one cell is cancerous and the other not, it is unclear if any conclusions can be made related to cell-type.

We corrected this to ‘*cell line*-dependent manner’.

HSPB5 is significantly up-regulated in U2OS ATM KO but not in HEK293T CPT treated cells. Does HSPB5 over-expression also prevent aggregation in HEK293T CPT treated cells? This would help to determine the breath of action of HSPB5 in combating genotoxic stress related protein aggregation.

We tested the overexpression of HSPB5 after CPT treatment in U2OS and HEK293 cells. Whereas it clearly suppressed the increased aggregation in U2OS cells (Figure 6H, I) it did not do so in HEK293 cells (Figure 6 —figure supplement 1G). We rephrased our conclusions and added a statement that provides a possible explanation (lines 490-500). As this reviewer already pointed out, the low levels of HSPB5 in HEK293 cells may be the culprit. In cells, small heat shock proteins such as HSBP5 operate together with other small heat shock proteins, and rely on other chaperone systems as well. It might be that cells (such as HEK293 cells) with low levels of small heat shock proteins are not “wired” to support excess HSPB5 function, and therefore its overexpression is unable to counteract genotoxic stress-induced aggregation. We discuss this in the Discussion.

Additional experiments to strengthen the science:I recommend comparing CPT treatment in U2OS versus HEK-293T cells and ATM inhibition by drug treatment versus CPT drug treatment in one cell type. It would be interesting to see the results related to SG proteins, heat-sensitive and enrichment for constituents of disease-associated protein aggregates for the ATM KO aggregation set.

We fully agree and now added these data. We also performed the downstream analyses again with the additional datasets. Importantly, these new datasets provide further support for almost all of the conclusions in the original manuscript.

Also it would be important to show that HSPB5 overexpression alleviates protein aggregation in HEK293T CPT treated cells.

We tested the effect of HSPB5 overexpression in HEK293T cells, and found that it does not prevent aggregation after CPT treatment (Figure 6 —figure supplement 1G). We discuss this in more detail in the Discussion.